# VARIATIONAL INFORMATION PURSUIT FOR INTERPRETABLE PREDICTIONS

**Aditya Chattopadhyay, Kwan Ho Ryan Chan, Benjamin D. Haeffele, Donald Geman**
Johns Hopkins University, USA, {`achatto1,kchan49,bhaeffele,geman`}`@jhu.edu`

**René Vidal**
University of Pennsylvania, USA, `vidalr@seas.upenn.edu`

## ABSTRACT

There is a growing interest in the machine learning community in developing predictive algorithms that are *interpretable by design*. To this end, recent work proposes to sequentially ask interpretable queries about data until a high confidence prediction can be made based on the answers obtained (the history). To promote short query-answer chains, a greedy procedure called Information Pursuit (IP) is used, which adaptively chooses queries in order of information gain. Generative models are employed to learn the distribution of query-answers and labels, which is in turn used to estimate the most informative query. However, learning and inference with a full generative model of the data is often intractable for complex tasks. In this work, we propose Variational Information Pursuit (V-IP), a variational characterization of IP which bypasses the need to learn generative models. V-IP is based on finding a query selection strategy and a classifier that minimize the expected cross-entropy between true and predicted labels. We prove that the IP strategy is the optimal solution to this problem. Therefore, instead of learning generative models, we can use our optimal strategy to directly pick the most informative query given any history. We then develop a practical algorithm by defining a finite-dimensional parameterization of our strategy and classifier using deep networks and train them end-to-end using our objective. Empirically, V-IP is 10-100x faster than IP on different Vision and NLP tasks with competitive performance. Moreover, V-IP finds much shorter query chains when compared to reinforcement learning which is typically used in sequential-decision-making problems. Finally, we demonstrate the utility of V-IP on challenging tasks like medical diagnosis where the performance is far superior to the generative modeling approach.

## 1 INTRODUCTION

Suppose a doctor diagnoses a patient with a particular disease. One would want to know not only the disease but also an evidential explanation of the diagnosis in terms of clinical test results, physiological data, or symptoms experienced by the patient. For practical applications, machine learning methods require an emphasis not only on metrics such as generalization and scalability but also on criteria such as interpretability and transparency. With the advent of deep learning methods over traditionally interpretable methods such as decision trees or logistic regression, the ability to perform complex tasks such as large-scale image classification now often implies a sacrifice in interpretability. However, interpretability is important in unveiling potential biases for users with different backgrounds (Yu, 2018) or gaining users' trust.

Most of the prominent work in machine learning that addresses this question of interpretability is based on post hoc analysis of a trained deep network's decisions (Simonyan et al., 2013; Ribeiro et al., 2016; Shrikumar et al., 2017; Zeiler & Fergus, 2014; Selvaraju et al., 2017; Smilkov et al., 2017; Chattopadhyay et al., 2019; Lundberg & Lee, 2017). These methods typically assign *importance scores* to different features used in a model's decision by measuring the sensitivity of the model output to these features. However, explanations in terms of importance scores of raw features might not always be as desirable as a description of the reasoning process behind a model's decision. Moreover, there are rarely any guarantees for the reliability of these post hoc explanations to faithfully represent the model's decision-making process (Koh et al., 2020). Consequently, post hoc interpretability has been widely criticized (Adebayo et al., 2018; Kindermans et al., 2019; Rudin,

2019; Slack et al., 2020; Shah et al., 2021; Yang & Kim, 2019) and there is a need to shift towards ML algorithms that are *interpretable by design*.

An interesting framework for making interpretable predictions was recently introduced by Chattopadhyay et al. (2022). The authors propose the concept of an *interpretable query set $Q$*, a set of user-defined and task-specific functions $q : \mathcal{X} \to \mathcal{A}$, which map a data point in $\mathcal{X}$ to an answer in $\mathcal{A}$, each having a clear interpretation to the end-user. For instance, a plausible query set for identifying bird species might involve querying beak shape, head colour, and other visual attributes of birds.

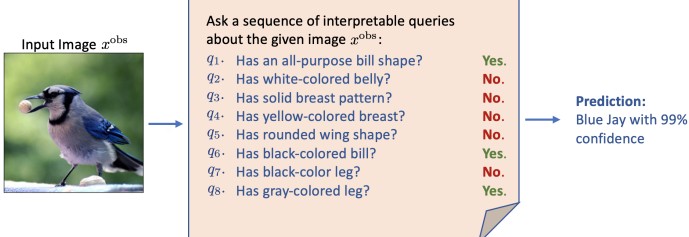

Figure 1: Illustration of the framework on a bird classification task. The query set consists of questions about the presence or absence of different visual attributes of birds. Given an image $x^{\text{obs}}$, a sequence of interpretable queries is asked about the image until a prediction can be made with high confidence. The choice of each query depends on the query-answers observed so far.

Given a query set, their method sequentially asks queries about $X$ until the answers obtained are sufficient for predicting the label/hypothesis $Y$ with high confidence. Notably, as the final prediction is solely a function of this sequence of query-answer pairs, these pairs provide a complete explanation for the prediction. Figure 1 illustrates the framework on a bird classification task.

To obtain short explanations (short query-answer chains), the authors propose to use a greedy procedure called Information Pursuit (IP), which was first introduced in Geman & Jedynak (1996). Given any input $x^{\text{obs}}$, IP sequentially chooses the query which has the largest mutual information about the label/hypothesis $Y$ given the history of query-answers obtained so far. To compute this mutual information criteria, a generative model is first trained to learn the joint distribution between all query-answers $q(X)$ and $Y$; in particular, Variational Autoencoders (VAEs) (Kingma & Welling, 2013) are employed. This learnt VAE is then used to construct Markov Chain Monte Carlo (MCMC) estimates for mutual information via MCMC sampling. Unfortunately, the computational costs of MCMC sampling coupled with the challenges of learning accurate generative models that enable fast inference limit the application of this framework to simple tasks. As an example, classifying MNIST digits using $3 \times 3$ overlapping patches as queries[1] with this approach would take weeks!

In this paper, we question the need to learn a full generative model between all query-answers $q(X)$ and $Y$ given that at each iteration IP is only interested in finding the most informative query given the history. More specifically, we present a variational charaterization of IP which is based on the observation that, given any history, the query $q^*$, whose answer minimizes the KL-divergence between the label distribution $P(Y \mid X)$ and the posterior $P(Y \mid q^*(X), \text{history})$, will be the most informative query as required by IP. As a result, we propose to minimize this KL-divergence term in expectation (over randomization of histories) by optimizing over querier functions, which pick a query from $Q$ given history, parameterized by deep networks. The optimal querier would then learn to directly pick the most informative query given any history, thus bypassing the need for explicitly computing mutual information using generative models. Through extensive experiments, we show that the proposed method is not only faster (since MCMC sampling methods are no longer needed for inference), but also achieves competitive performance when compared with the generative modeling approach and also outperforms other state-of-the-art sequential-decision-making methods.

**Paper Contributions.** (1) We present a variational characterization of IP, termed Variational-IP or V-IP, and show that the solution to the V-IP objective is exactly the IP strategy. (2) We present a practical algorithm for optimizing this objective using deep networks. (3) Empirically, we show that V-IP achieves competitive performance with the generative modelling approach on various computer vision and NLP tasks with a much faster inference time. (4) Finally, we also compare our approach to Reinforcement Learning (RL) approaches used in sequential-decision making areas like Hard Attention (Mnih et al., 2014) and Symptom Checking (Peng et al., 2018), where the objective is to learn a policy which adaptively chooses a fixed number of queries, one at a time, such that an accurate prediction can be made. In all experiments, V-IP is superior to RL methods.

---

[1] Each patch query asks about the pixel intensities observed in that patch for $x^{\text{obs}}$.

## 2 RELATED WORK

**Interpretability in Machine Learning.** These works can be broadly classified into two main categories: (i) *post-hoc interpretability*, and (ii) algorithms that are *interpretable by design*. A large number of papers in this area are devoted to *post-hoc* interpretability. However, as stated in the Introduction, the reliability of these methods have recently been called into question (Adebayo et al., 2018; Yang & Kim, 2019; Kindermans et al., 2019; Shah et al., 2021; Slack et al., 2020; Rudin, 2019; Koh et al., 2020; Subramanya et al., 2019). Consequently, recent works have focused on developing ML algorithms that are *interpretable by design*. Several of these works aim at learning deep networks via regularization such that it can be approximated by a decision tree (Wu et al., 2021) or locally by a linear network (Bohle et al., 2021; Alvarez Melis & Jaakkola, 2018). However, the framework of Chattopadhyay et al. (2022) produces predictions that are completely explained by interpretable query-chains and is not merely an approximation to an interpretable model like decision trees. Another line of work tries to learn latent semantic concepts or prototypes from data and subsequently base the final prediction on these learnt concepts (Sarkar et al., 2022; Nauta et al., 2021; Donnelly et al., 2022; Li et al., 2018; Yeh et al., 2020). However, there is no guarantee that these learnt concepts would be interpretable to the user or align with the user's requirement. In sharp contrast, allowing the user to define an interpretable query set in (Chattopadhyay et al., 2022) guarantees by construction that the resulting query-chain explanations would be interpretable and useful.

**Sequential Decision-Making.** An alternative approach to learning short query-chains is to use methods for sequential decision learning. These algorithms can be used for making interpretable decisions by sequentially deciding "what to query next?" in order to predict $Y$ as quickly as possible. Mnih et al. (2014) introduced a reinforcement-learning (RL) algorithm to sequentially observe an image through glimpses (small patches) and predict the label, and called their approach Hard Attention. Rangrej & Clark (2021) introduced a probabilistic model for Hard Attention which is similar to the IP algorithm. More specifically, they propose to learn a partial-VAE Ma et al. (2018) to directly learn the distribution of images given partially observed pixels. This VAE is then used to select glimpses in order of information gain, as in IP. In another work, Peng et al. (2018) introduced an RL-based framework to sequentially query patient symptoms for fast diagnosis. In §4, we compare V-IP with prior works in this area and show that in almost all cases our method requires a smaller number of queries to achieve the same level of accuracy. We conjecture that the superiority of V-IP over RL-based methods is because the V-IP optimization is not plagued by sparse rewards over long trajectories, for example, a positive reward for correct prediction after a large number of symptom queries as in Peng et al. (2018). Instead, Deep V-IP can be abstractly thought of as, given history, $s$, choosing a query $q$ (the action) and receiving $D_{\mathrm{KL}}(P(Y \mid x) \mid\mid P(Y \mid q(x), s))$ as an immediate reward. A more rigorous comparison of the two approaches would be an interesting future work.

## 3 METHODS

### 3.1 G-IP: INFORMATION PURSUIT VIA GENERATIVE MODELS AND MCMC

Let $X : \Omega \to \mathcal{X}$ and $Y : \Omega \to \mathcal{Y}$ be random variables representing the input data and corresponding labels/output. We use capital letters for random variables and small letters for their realizations. $\Omega$ is the underlying sample space on which all random variables are defined. Let $P(Y \mid X)$ denote the ground truth conditional distribution of $Y$ given data $X$. Let $Q$ be a set of task-specific, user-defined, interpretable functions of data, $q : \mathcal{X} \to \mathcal{A}$, where $q(x) \in \mathcal{A}$ is the answer to query $q \in Q$ evaluated at $x \in \mathcal{X}$. We assume that $Q$ is sufficient for solving the task, i.e., we assume that $\forall (x, y) \in \mathcal{X} \times \mathcal{Y}$

$$P(y \mid x) = P(y \mid \{x' \in \mathcal{X} : q(x') = q(x) \,\forall q \in Q\}). \tag{1}$$

In other words $Q(X) := \{q(X) : q \in Q\}$ is a sufficient statistic for $Y$.

The Information Pursuit (IP) algorithm (Geman & Jedynak, 1996) proceeds as follows; given a data-point $x^{\mathrm{obs}}$, a sequence of most informative queries is selected as

$$q_1 = \mathrm{IP}(\emptyset) = \arg\max_{q \in Q} I(q(X); Y); \tag{2}$$

$$q_{k+1} = \mathrm{IP}(\{q_i, q_i(x^{\mathrm{obs}})\}_{1:k}) = \arg\max_{q \in Q} I(q(X); Y \mid q_{1:k}(x^{\mathrm{obs}})).$$

Here $q_{k+1} \in Q$ refers to the new query selected by IP at step $k+1$, based on the history (denoted as $q_{1:k}(x^{\text{obs}}))$[2], and $q_{k+1}(x^{\text{obs}})$ indicates the corresponding answer. The algorithm terminates after $L$ queries, where $L$ depends on the data point $x^{\text{obs}}$, if all remaining queries are nearly uninformative, that is, $\forall q \in Q \; I(q(X); Y \mid q_{1:L})) \approx 0$. The symbol $I$ denotes mutual information. Evidently equation 2 requires estimating the query with maximum mutual information with $Y$ based on history. One approach to carrying out IP is by first learning the distribution $P(Q(X), Y)$ from data using generative models and then using MCMC sampling to estimate the mutual information terms[3]. However, learning generative models for distributions with high-dimensional support is challenging, and performing multiple iterations of MCMC sampling can likewise be computationally demanding. To address this challenge, in the nest subsection we propose a variational characterization of IP that completely bypasses the need to learn and sample from complex generative models.

## 3.2 V-IP: A Variational Characterization of Information Pursuit

We begin this section by describing our variational characterization of IP. The proposed approach is motivated by the fact that generative models are only a means to an end; what we need is the function, that we call *querier*, that maps the histories observed, $\{q_i, q_i(x^{\text{obs}})\}_{1:k}$, to the most informative next query $q_{k+1} \in Q$. It turns out that this most informative query is exactly the query $q^*$ whose answer will minimize the KL divergence between the conditional label distribution $P(Y \mid X)$ and

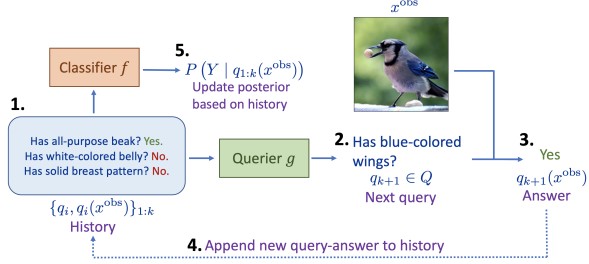

Figure 2: **Overview.** Interpretable predictions using V-IP.

the posterior $P(Y \mid q^*(X), \{q_i(x^{\text{obs}})\}_{1:k})$. Based on this insight, is it possible to define an optimization problem to directly learn this querier function? This requires a few ingredients,

- First, we need to learn a querier that, given any possible history one might encounter during IP, chooses the next most informative query. One possible strategy for this is to minimize the KL divergence objective in expectation over random histories of query-answer pairs.
- The posterior $P(Y \mid q^*(X), \{q_i(x^{\text{obs}})\}_{1:k})$ depends on the data distribution and is typically unknown. Thus, we need to estimate this using probabilistic classifiers. A possible solution is to jointly optimize this expected KL divergence over both querier and classifier functions.

This leads to the following variational characterization of IP which will allow us to avoid generative models. Let $\mathbb{K}(x)$ be the set of all finite-length query-answer pairs of the form $(\{q_1, q_1(x)\}, ..., \{q_m, q_m(x)\})$, generated using queries from $Q$ evaluated on any $x \in \mathcal{X}$. We then define $\mathbb{K} := \cup_{x \in \mathcal{X}} \mathbb{K}(x)$ and denote elements of $\mathbb{K}$ as "histories". We define a classifier $f : \mathbb{K} \to \mathcal{P}_{\mathcal{Y}}$ as a function which maps arbitrary query-answer sequences to a distribution over $\mathcal{Y}$. We define a querier $g : \mathbb{K} \to Q$ as a function which maps arbitrary query-answer sequences to a query $q \in Q$. The variational objective for IP is given by the following functional optimization problem,

$$
\begin{aligned}
\min_{f,g} \quad & \mathbb{E}_{X,S} \left[ D_{\text{KL}} \left( P(Y \mid X) \,\|\, \hat{P}(Y \mid q(X), S) \right) \right] \\
\text{where} \quad & q := g(S) \in Q \\
& \hat{P}(Y \mid q(X), S) := f(\{q, q(X)\} \cup S),
\end{aligned}
\tag{V-IP}
$$

and the minimum is taken over all possible mappings $f$ (classifier) and $g$ (querier). Here, $S$ is a random set of query-answer pairs taking values in $\mathbb{K}$[4]. Given $S = s$ and $X = x^{\text{obs}}$, the querier $g$ chooses a query $q \in Q$, evaluates it on $x^{\text{obs}}$ and passes the pair $\{q, q(x^{\text{obs}})\}$ to the classifier. The classifier $f$ then makes a prediction based on $s$ appended with this additional pair $\{q, q(x^{\text{obs}})\}$.

---

[2]Conditioning on $q_{1:k}(x^{\text{obs}})$ is to be understood as conditioning on the event $\{x' \in \mathcal{X} \mid \{q_i, q_i(x^{\text{obs}})\}_{1:k} = \{q_i, q_i(x')\}_{1:k}\}$

[3]Since mutual information requires computing expectation over density ratios which is still intractable despite having learnt a generative model.

[4]Throughout this paper whenever we condition on $S = s$, we mean conditioning on the event of all data $x' \in \mathcal{X}$ which share the same answers to queries as in $s$.

Let $(f^*, g^*)$ be an optimal solution to V-IP. The querier $g^*$ will be the IP strategy with the requirement that the distribution of $S$, denoted as $P_S$, in V-IP is chosen such that the histories observed while carrying out IP must have positive probability mass under $P_S$. Thus, given a data-point $x^{\text{obs}}$,

$$q_1 = g^*(\emptyset) = \underset{q \in Q}{\arg\max}\, I(q(X); Y);$$

$$q_{k+1} = g^*(\{q_i, q_i(x^{\text{obs}})\}_{1:k}) = \underset{q \in Q}{\arg\max}\, I(q(X); Y \mid q_{1:k}(x^{\text{obs}})). \tag{3}$$

The above sequential procedure is illustrated in Figure 2. As before $\{q_i, q_i(x^{\text{obs}})\}_{1:k}$ is referred to as the history observed after $k$ queries and is a realization of $S$. This is formalized in the following proposition whose proof can be found in Appendix A.

**Proposition 1.** *Let $(f^*, g^*)$ be an optimal solution to V-IP. For any realization $S = s$ such that $P(S = s) > 0$, define the optimization problem:*

$$\max_{\tilde{P} \in \mathcal{P}_{\mathcal{Y}}, q \in Q} I(q(X); Y \mid s) - \mathbb{E}_{X \mid s}\left[ D_{\text{KL}}\left( P(Y \mid q(X), s) \,\|\, \tilde{P}(Y \mid q(X), s) \right) \right]. \tag{4}$$

*Then there exists an optimal solution $(\tilde{P}_s^*, q_s^*)$ to the above objective such that $q_s^* = g^*(s)$ and $\tilde{P}_s^* = f^*(\{q_s^*, q_s^*(X)\} \cup s)$.*

Thus, at the optima, the KL divergence term in equation 4 would be 0 and $g^*$ would pick the most informative query for any given subset of query-answer pairs $S = s$ as is presented in equation 3.

Theoretical guarantees aside, solving the optimization problem defined in V-IP is challenging since functional optimization over all possible classifier and querier mappings is intractable. In the following subsection, we present a practical algorithm for approximately solving this V-IP objective.

### 3.3 V-IP WITH DEEP NETWORKS

Instead of optimizing $f$ and $g$ over the intractable function space, we parameterize them using deep networks with weights $\theta$ and $\eta$ respectively. Our practical version of V-IP, termed as Deep V-IP, is as follows,

$$\min_{\theta, \eta} \quad \mathbb{E}_{X,S}[D_{\text{KL}}(P(Y \mid X) \,\|\, P_\theta(Y \mid q_\eta(X), S)]$$

$$\text{where} \quad q_\eta := g_\eta(S) \tag{Deep V-IP}$$

$$P_\theta(Y \mid q_\eta(X), S) := f_\theta(\{q_\eta, q_\eta(X)\} \cup S).$$

Note that all we have done is replace arbitrary functions $f$ and $g$ in V-IP by deep networks parameterized by $\theta$ and $\eta$. To find good solutions there are two key constraints. First, the architecture for the classifier $f_\theta$ and the querier $g_\eta$ functions need to be expressive enough to learn over an exponential (in $|Q|$) number of possible realizations of $S$. Second, we need to choose a sampling distribution $P_S$ for $S$. Notice, that for any reasonable-sized $Q$, there will be an exponentially large number of possible realizations of $S$. Ideally, we would like to choose a $P_S$ that assigns positive mass only to histories observed during the exact IP procedure, however this is like the "chicken-and-egg" dilemma. We now briefly discuss the architectures used for optimizing the Deep V-IP objective, followed by an exposition on the sampling distribution for $S$.

**Architectures.** The architecture for both the querier and classifier networks (described in more detail in Appendix C) are chosen in such a way that they can operate on arbitrary length query-answer pairs. There can be several choices for this. In this paper, we primarily use masking where the deep networks operate on fixed size inputs (the answers to all queries $q \in Q$ evaluated on input $x$) with the unobserved query-answers masked out. We also experiment with set-based deep architectures proposed in Ma et al. (2018). We show by ablation studies in Appendix E that the masking-based architecture performs better. In practice, $q_\eta = \texttt{argmax}(g_\eta(S))$, where $g_\eta(S) \in \mathbb{R}^{|Q|}$ is the output of the querier network, which assigns a score to every query in $Q$ and $\texttt{argmax}$ computes an 1-hot indicator of the $\texttt{max}$ element index of its input vector. To ensure differentiability through $\texttt{argmax}$ we use the straight-through softmax estimator (Paulus et al., 2020) which is described in detail in Appendix D. Finally, $P_\theta(Y \mid q_\eta(X), S)$ is the output of Softmax applied to the last layer of $f_\theta$.

**Sampling of $S$.** Choosing a sampling distribution that only has positive mass on histories observed during exact IP is like the "chicken-and-egg" dilemma. A simple alternative is to consider a distribution that assigns a positive mass to all possible sequences of query-answer pairs from $\bar{\mathbb{K}}$. This

however would lead to slow convergence since the deep networks now have to learn to choose the most informative query given a large number query-answer pair subsets that would never be observed for any $x^{\text{obs}} \in \mathcal{X}$ if one could do exact IP. To remedy this, we choose to adaptively bias our sampling distribution towards realizations of $S$ one would observe if they carried out equation 3 using the current estimate for querier in place of $g^*$. More concretely, we optimize Deep V-IP by sequentially biasing the sampling distribution as follows:

1. Initial Random Sampling: We choose an initial distribution $P_S^0$ which ensures all elements of $\bar{\mathbb{K}}$ have positive mass. We first sample $X \sim P_{\text{Data}}$. Then we sample $k \sim \text{Uniform}\{0, 1, ..., |Q|\}$ as the number of queries. Subsequently, $k$ queries from $Q$ are selected for $X$ uniformly at random.
2. Subsequent Biased Sampling: The distribution $P_S^{j+1}$ is obtained by using the solution querier $g_{\eta^j}$ to V-IP using $P_S^j$ as the sampling distribution. In particular, we first sample $X \sim P_{\text{Data}}$ and $k \sim \text{Uniform}\{0, 1, ..., |Q|\}$ as before. Subsequently, we find the first $k$ query-answer pairs for this sampled $X$ using equation 3 with $g_{\eta^j}$ as our querier.

Notice that, the empty set $\emptyset$, corresponding to empty history, would have positive probability under any $P_S^j$ and hence the querier would eventually learn to pick the most informative first query. Subsequent sequential optimization would aim at choosing the most informative second query and so on, assuming our architectures are expressive enough. In practice, we optimize with random sampling of $S$ using stochastic gradients for numerous epochs. We then take the solution $g_{\eta^0}$ and fine-tune it with biased sampling strategies, each time optimizing using a single batch and consequently changing the sampling strategy according to the updated querier. Refer Appendix E for ablation studies on the effectiveness of the biased sampling strategy for $S$.

**Stopping Criterion.** There are two possible choices; (i) *Fixed budget*: Following prior work in sequential active testing (Ma et al., 2018; Rangrej & Clark, 2021), we stop asking queries after a fixed number of iterations. (ii) *Variable query-lengths*: Different data-points might need different number of queries to make confident predictions. For supervised learning tasks, where $Y$ is "almost" a deterministic function of $X$, that is, $\max_Y P(Y \mid X) \approx 1$, for any given $x^{\text{obs}} \in \mathcal{X}$, we terminate after $L$ steps if $\max_Y P(Y \mid q_{1:L}(x^{\text{obs}})) \geq 1 - \epsilon$, where $\epsilon$ is a hyperparameter. This is termed as the "MAP criterion". For tasks where $Y$ is more ambiguous and not a deterministic function of $X$, we choose to terminate once the posterior is "stable" for a pre-defined number of steps. This stability is measured by the difference between the two consecutive posterior entropies, $H\left(Y \mid q_{1:k}(x^{\text{obs}})\right) - H\left(Y \mid q_{1:k+1}(x^{\text{obs}})\right) \leq \epsilon$. This criterion, termed as the "stability criterion", is an unbiased estimate of the mutual information-based stopping criteria used in Chattopadhyay et al. (2022).

**Qualitative differences between Generative-IP and Variational-IP.** We will refer to the generative approach for carrying out IP as described in Chattopadhyay et al. (2022) as Generative-IP or G-IP. The difference between G-IP and V-IP is similar in spirit to that of generative versus discriminative modelling in classification problems (Ng & Jordan, 2001). We conjecture that, when the data distribution agrees with the modelling assumptions made by the generative model, for example, conditional independence of query answers given $Y$, and the dataset size is "small," then G-IP would obtain better results than V-IP since there are not enough datapoints for learning competitive querier and classifier networks. We thus expect the gains of V-IP to be most evident on datasets where learning a good generative model is difficult.

## 4 EXPERIMENTS

In this section, through extensive experiments, we evaluate the effectiveness of the proposed method. We describe the query set used for each dataset in Table 1, with more details in Appendix C. The choice of query sets for each dataset was made to make our approach comparable with prior work. We also complement the results presented here with more examples in the Appendix. Code is available at `https://github.com/ryanchankh/VariationalInformationPursuit`.

### 4.1 INTERPRETABLE PREDICTIONS USING V-IP

Basing predictions on an interpretable query set allows us to reason about the predictions in terms of the queries, which are compositions of elementary words, symbols or patterns. We will illustrate this by analyzing the query-answer chains uncovered by V-IP for different datasets. Figure 3a illustrates

Table 1: Descriptions of query set for different datasets.

| Dataset | Query Set $Q$ | Size $|Q|$ |
|---|---|---|
| CUB-200 (Wah et al., 2011) | indicator for the presence of different visual semantic attributes of birds | 312 |
| HuffingtonNews (Misra, 2018) | indicator for presence of words in article headline or description | 1000 |
| MNIST (LeCun et al., 1998), KMNIST (Clanuwat et al., 2018), Fashion-MNIST (Xiao et al., 2017b), | pixel intensities in $3 \times 3$ overlapping patches with stride 1 | 676 |
| CIFAR-{10,100} (Krizhevsky et al., 2009) | pixel intensities in $8 \times 8$ overlapping patches with stride 4 | 49 |
| SymCAT-200 (Peng et al., 2018) SymCAT-300 (Peng et al., 2018) SymCAT-400 (Peng et al., 2018) | indicator for the presence of different medical symptoms | 328 349 355 |
| MuZhi (Wei et al., 2018) Dxy (Xu et al., 2019) | ternary {Yes, No, Can't Say} queries for the presence of different medical symptoms | 66 41 |

the decision-making process for V-IP on an image of a dog from the CIFAR-10 dataset. A priori the model's belief is almost uniform for all the classes (second row, first column). The first query probes a patch near the centre of the image and observes the snout. Visually it looks similar to the left face of a cat, justifying the shift in the model's belief to label "cat" with some mass on the label "dog". Subsequent queries are aimed at distinguishing between these two possibilities. Finally, the model becomes more than $99\%$ confident that it is a "dog" once it spots the left ear. Figure 3b shows the query-chain for a "genital herpes" diagnosis of a synthetic patient from the SymCAT-200 dataset. The y-axis shows the query asked at each iteration with green indicating a "Yes" answer and red indicating a "No". Each row shows the model's current belief in the patient's disease. We begin with an initial symptom, "0: itching of skin", provided by the patient. The subsequent queries ask about different conditions of the skin. The diseases shown are the top-10 most probable out of 200. All of these diseases have skin-related symptoms. After discovering the patient has painful urination (query 11), V-IP zooms into two possibilities "Balanitis" and "Genital herpes". The subsequent queries rule out symptoms typically observed in patients with "Balanties" resulting in a $80\%$ confidence in the herpes disease. For our final example, we elucidate the results for a bird image from the CUB-200 dataset in Figure 3c. The colour scheme for the y-axis is the same as in Figure 3b, with the exception that, unlike the patient case, we do not bootstrap V-IP with an initial positive attribute of the word. Instead, the first query about bill shape is the most-informative query about the label $Y$ before any answer is observed. This is indicated with the grey "0:Init". All the top-10 most probable bird species in this context are seabirds, and have very similar visual characteristics to the true species, "Laysan Albatross". After $14$ queries V-IP figures out the true class with more than $99\%$ confidence. Thus, in all three case-studies, we see that V-IP makes transparent decisions, interpretable in terms of queries specified by the user. A limitation of this framework however is finding a good query set that is interpretable and allows for highly accurate predictions with short explanations (short query-answer chains). We discuss this further in Appendix §G.

## 4.2 QUANTITATIVE COMPARISON WITH PRIOR WORK

**Baselines.** We compare V-IP primarily to the generative modelling approach for IP, namely G-IP. We also compare to Reinforcement-Learning methods prevalent in other areas of sequential-decision making like Hard-Attention (Mnih et al., 2014; Rangrej & Clark, 2021) or Symptom Checking (Peng et al., 2018), which can be adapted for our purposes. In particular, we compare with the RAM (Mnih et al., 2014) and RAM+(Li et al., 2017) algorithms. In both methods, a policy is learnt using deep networks to select queries based on previous query-answers for a fixed number of iterations such that the expected cumulative reward is maximized. A classifier network is also trained simultaneously with the policy network to make accurate predictions. In RAM, this reward is just the negative cross-entropy loss between true and predicted labels at the last step. In RAM+, this reward is the cumulative sum of the negative cross-entropy loss at each step. We also compare our method with the "Random" strategy where successive queries are

Table 2: AUC values for test accuracy vs. explanation length curves for different datasets. To account for different query set sizes across datasets, we normalize the scores by $|Q|$. Refer to Table 4 in Appendix §F for extended results with standard deviations for other methods (averaged over 5 runs).

| Dataset | Random | RAM | RAM+ | G-IP | V-IP (Ours) |
|---|---|---|---|---|---|
| CUB | 0.557 | 0.662 | 0.695 | **0.736** | $0.716 \pm 0.008$ |
| HuffingtonNews | 0.423 | 0.389 | 0.431 | **0.691** | $0.680 \pm 0.002$ |
| MNIST | 0.868 | 0.916 | 0.920 | **0.964** | $0.956 \pm 0.002$ |
| KMNIST | 0.775 | 0.832 | 0.841 | 0.872 | **$0.911 \pm 0.008$** |
| Fashion-MNIST | 0.735 | 0.770 | 0.804 | 0.831 | **$0.849 \pm 0.010$** |

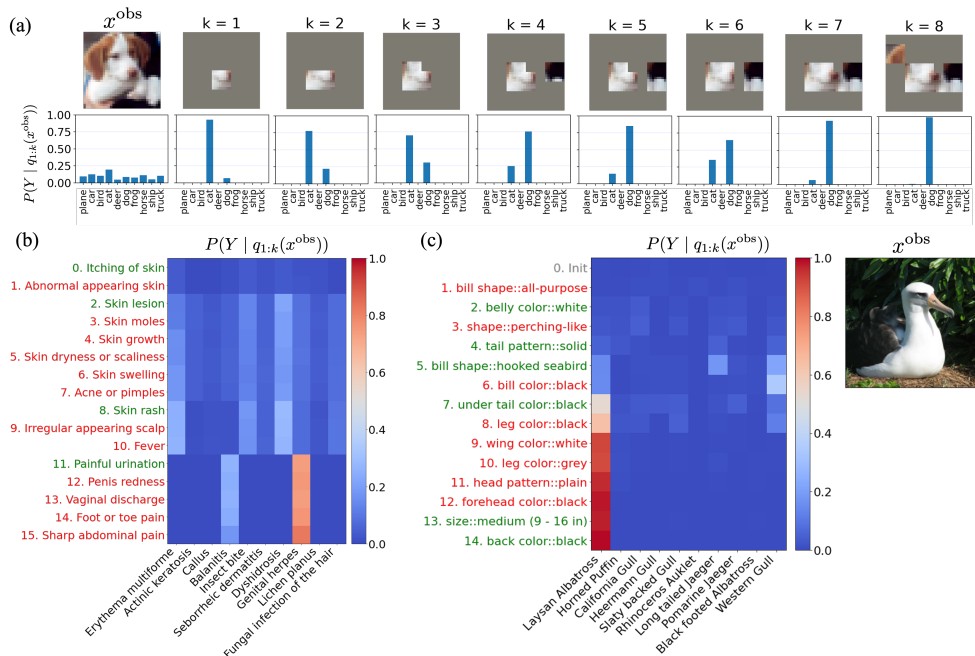

Figure 3: **(a) V-IP on CIFAR-10.** In column 1 row 1, we show the observed image $x^{\text{obs}}$, and in column 1 row 2, we show the label distribution before V-IP observes any patches. In the subsequent columns, row 1 indicates the patches revealed (the history) so far and row 2 shows the corresponding posterior over the labels given this history. **(b) V-IP on SymCAT-200.** Each row in the heatmap shows the posterior of the disease labels given history. We show the top-10 most probable diseases out of 200. The y-axis indicates the corresponding symptom queried in each iteration by V-IP. We use the colour scheme that red denotes a "No" answer while green denotes a "Yes" answer. **(c) V-IP on CUB-200.** The observed image $x^{\text{obs}}$ is shown on the right. The heatmap shows the posterior, similar to (b).

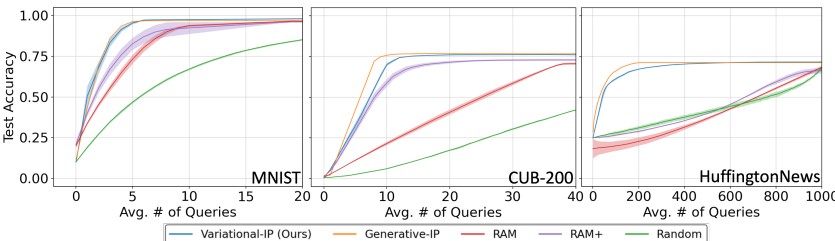

Figure 4: Tradeoff between accuracy and explanation length (avg. number of queries). The curves were generated by varying the $\epsilon$ parameter in the stopping criteria (see §3.3). For MNIST & HuffingtonNews we use the "MAP criterion", whereas for CUB-200 we use the posterior "stability criterion". Reported curves are averaged over 5 runs with shaded region denoting the standard deviation. More results in Appx. F.

chosen randomly independent of the history observed so far. The predictions given history are still made using the V-IP classifier.

We first show results on the simple datasets used in Chattopadhyay et al. (2022) comparing with our own implementation of G-IP, RAM and RAM+. V-IP is competitive with G-IP in terms of performance but far more efficient in terms of speed of inference. In all datasets, V-IP outperforms the RL-based methods. Subsequently, we present results on more complex tasks like RGB image classification. On these datasets, V-IP achieves a higher accuracy given a fixed budget of queries compared with prior work.

**Comparisons on Simple Tasks.** Concise explanations are always preferred due to their simplicity. In Figure 4 we plot the trade-off between accuracy and explanation length obtained by various methods. V-IP is competitive with G-IP and obtains far shorter explanations than the RL-based methods to obtain the same test accuracy. This trade-off is quantified using the Area Under the Curve (AUC) metric in Table 2. Notice on the HuffingtonNews dataset the RL methods struggle

to perform better than even Random. This is potentially due to the fact that in these RL methods, the classifier is trained jointly with the policy which likely affects its performance when the action-space is large ($|Q|$ = 1000). On the other hand, the random strategy learns its classifier by training on random sequences of query-answer pairs. This agrees with findings in Rangrej & Clark (2021).

While V-IP performs competitive with the generative approach the biggest gain is in terms of computational cost of inference. Once trained, inference in V-IP, that is computing the most-informative query, is akin to a forward pass through a deep network and is potentially $\mathcal{O}(1)^5$. On the other hand, the per-iteration cost in G-IP is $\mathcal{O}(N + |Q|m)$, where $N$ is the number of MCMC iterations employed and $m$ is the cardinality of the space $q(X) \times Y$. As an example, on the same GPU server, G-IP takes about 47 seconds per iteration on MNIST whereas V-IP requires just 0.11s, an approximately $400\times$ speedup! Note that the inference cost is the same for V-IP and the RL methods since all of them train a querier/policy function to choose the next query.

**Comparisons on Complex Tasks.** We now move on to more complex datasets where the gains of V-IP are more evident. First, we consider the task of natural image classification and show results on the CIFAR-$\{10, 100\}$ datasets. For G-IP on these datasets, we refer to the Probabilistic HardAttn model introduced in Rangrej & Clark (2021) which proposes to learn a partial-VAE model (Ma et al., 2018) for images and then do inference using this model to compute the most informative query. Figures 5 a & b show the accuracy vs. number of queries curves for different methods. V-IP clearly outperforms all baselines on both datasets.

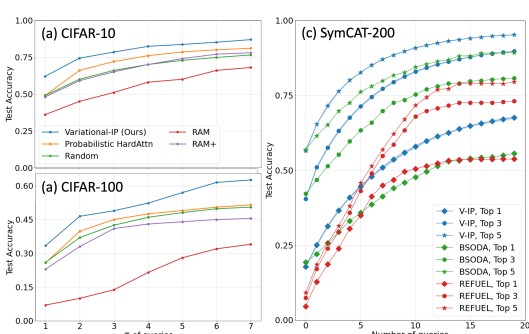

Figure 5: Tradeoff between accuracy and number of queries asked (fixed budged stopping criterion) for the CIFAR datasets (a & b) and the SymCAT-200 dataset (c). Curves for baselines adapted from Rangrej & Clark (2021) for (a) & (b) and from He et al. (2022) for (c).

Next, we consider the task of medical diagnosis by querying symptoms of the patient. We show results on the popular SymCAT dataset along with comparisons with prior work in Figure 5c. The plot clearly shows that V-IP achieves a much higher accuracy given a fixed budget of queries.

For G-IP on this task, we refer to the BSODA framework introduced in He et al. (2022) which is again based on partial-VAEs. REFUEL (Peng et al., 2018) is a state-of-the-art RL-based method for this task akin to the RAM technique used in Hard-Attention literature. The classification accuracy for these different methods on all medical datasets are summarized in Table 3. Numbers for baselines are taken from Nesterov et al. (2022) since we used

Table 3: Test accuracy obtained on different symptom checking datasets after asking 20 queries.

| Dataset | BSODA | REFUEL | V-IP (Ours) |
|---|---|---|---|
| SymCAT-200 | 0.556 | 0.548 | **0.681** $\pm$ 0.004 |
| SymCAT-300 | 0.475 | 0.482 | **0.599** $\pm$ 0.007 |
| SymCAT-400 | 0.446 | 0.438 | **0.513** $\pm$ 0.009 |
| MuZhi | **0.726** | 0.718 | 0.706 $\pm$ 0.011 |
| Dxy | **0.811** | 0.757 | 0.806 $\pm$ 0.019 |

their released versions of these datasets[6]. As conjectured in §3.3, MuZhi and Dxy are small-scale datasets with about 500 training samples thus approaches based on generative models, like BSODA, are able to perform slightly better than V-IP.

## 5 CONCLUSION

IP was recently used to construct interpretable predictions by composing interpretable queries from a user-defined query set. The framework however required generative models which limited its application to simple tasks. Here, we have introduced a variational characterization of IP which does away with generative models and tries to directly optimize a KL-divergence based objective to find the most informative query, as required by IP, in each iteration. Through qualitative and quantitative experiments we show the effectiveness of the proposed method.

---

[5] For simplicity we consider unit cost for any operation that was computed in a batch concurrently on a GPU.

[6] https://github.com/SympCheck/NeuralSymptomChecker

ACKNOWLEDGMENTS

This research was supported by the Army Research Office under the Multidisciplinary University Research Initiative contract W911NF-17-1-0304, the NSF grant 2031985 and by Simons Foundation Mathematical and Scientific Foundations of Deep Learning (MoDL) grant 135615. Moreover, the authors acknowledge support from the National Science Foundation Graduate Research Fellowship Program under Grant No. DGE2139757. Any opinions, findings, and conclusions or recommendations expressed in this material are those of the author(s) and do not necessarily reflect the views of the National Science Foundation.

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

APPENDIX

## A  PROOF OF PROPOSITION 1

Before proceeding to the proof we will prove the following lemma,

**Lemma 1.** *Let $Q$ be an user-defined query set and $\mathbb{P}(Y)$ be the set of all possible distributions on $Y$. Then, for any realization $S = s$, the following holds true:*

$$
\begin{aligned}
& \min_{\tilde{P} \in \mathcal{P}_\mathcal{Y}, q \in Q} \mathbb{E}_{X|s} \left[ D_{\mathrm{KL}}(P(Y \mid X) \mid\mid \tilde{P}(Y \mid q(X), s) \right] \\
& \equiv \max_{\tilde{P} \in \mathcal{P}_\mathcal{Y}, q \in Q} \left[ I(q(X); Y \mid s) - \mathbb{E}_{X|s}[D_{\mathrm{KL}}(P(Y \mid q(X), s) \mid\mid \tilde{P}(Y \mid q(X), s))] \right]
\end{aligned}
\tag{5}
$$

*Proof.* Using information-theoretic properties of the KL-divergence we have the following set of equalities.

$$
\begin{aligned}
& \min_{\tilde{P} \in \mathcal{P}_\mathcal{Y}, q \in Q} \mathbb{E}_{X|s} \left[ D_{\mathrm{KL}}(P(Y \mid X) \mid\mid \tilde{P}(Y \mid q(X), s) \right] \\
& = \min_{\tilde{P} \in \mathcal{P}_\mathcal{Y}, q \in Q} \mathbb{E}_{X|s} \left[ D_{\mathrm{KL}}(P(Y \mid X, q(X), s) \mid\mid \tilde{P}(Y \mid q(X), s) \right] \\
& = \min_{\tilde{P} \in \mathcal{P}_\mathcal{Y}, q \in Q} \mathbb{E}_{X|s} \left[ \sum_Y P(Y \mid X, q(X), s) \log \frac{P(Y \mid X, q(X), s)}{\tilde{P}(Y \mid q(X), s)} \right] \\
& = \min_{\tilde{P} \in \mathcal{P}_\mathcal{Y}, q \in Q} \mathbb{E}_{X|s} \left[ \sum_Y P(Y \mid X, q(X), s) \log \frac{P(X, Y \mid q(X), s)}{\tilde{P}(Y \mid q(X), s) P(X \mid q(X), s)} \right] \\
& = \min_{\tilde{P} \in \mathcal{P}_\mathcal{Y}, q \in Q} \mathbb{E}_{X,Y|s} \left[ \log \frac{P(X, Y \mid q(X), s)}{P(Y \mid q(X), s) P(X \mid q(X), s)} \right] + \mathbb{E}_{X,Y|s} \left[ \log \frac{P(Y \mid q(X), s)}{\tilde{P}(Y \mid q(X), s)} \right] \\
& = \min_{\tilde{P} \in \mathcal{P}_\mathcal{Y}, q \in Q} I(X; Y \mid q(X), s) + \mathbb{E}_{X|s}[D_{\mathrm{KL}}(P(Y \mid q(X), s) \mid\mid \tilde{P}(Y \mid q(X), s))]
\end{aligned}
\tag{6}
$$

In the first equality, assuming $P(X = x, S = s) > 0$,[7], we used the fact that given any $X = x$, the label $Y$ is independent of any query answer $q(X) = q(x)$ and event $\{S = s\}$. Thus, $P(Y \mid X = x) = P(Y \mid X = x, q(X) = q(x), S = s)$. In the fourth equality we multiplied the term inside the log by the identity $\frac{P(Y|q(X),s)}{P(Y|q(X),s)}$, where $P(Y \mid q(X), s)$ represents the true posterior of $Y$ given the query answer $q(X)$ and $S = s$.

Now observe that for any fixed $S = s$ and any $q \in Q$,

$$
\begin{aligned}
I(X, q(X); Y \mid s) & = I(X; Y \mid s) + I(q(X); Y \mid X, s) \\
& = I(X; Y \mid s)
\end{aligned}
\tag{7}
$$

The second equality is obtained by using the fact that $q(X)$ is a function of $X$.

Decomposing $I(X, q(X); Y \mid s)$ another way,

$$
I(X, q(X); Y \mid s) = I(q(X); Y \mid s) + I(X; Y \mid q(X), s)
\tag{8}
$$

From equation 7 and equation 8 we conclude that

$$
\min_{q \in Q} I(Y; X \mid q(X), s) \equiv \min_{q \in Q} -I(q(X); Y \mid s)
$$

Substituting the RHS in the above result in equation 6 we obtain the desired result.  □

---

[7]For any $x' \in \mathcal{X}$, if $P(X = x', S = s) = 0$, then $x'$ would not contribute to the expectation in the first equation and so we do not need consider this case.

**Proof of Proposition 1.**
Restating the objective from equation V-IP,

$$\min_{f,g} \quad \mathbb{E}_{X,S}\left[ D_{\mathrm{KL}}\left( P(Y \mid X) \parallel \hat{P}(Y \mid q(X), S) \right) \right]$$
$$\text{where} \quad q := g(S) \in Q$$
$$\hat{P}(Y \mid q(X), S) := f(\{q, q(X)\} \cup S),$$

Now, for any realization $S = s$, such that $P(S = s) > 0$, we have,

$$\min_{\tilde{P} \in \mathcal{P}_{\mathcal{Y}}, q \in Q} \mathbb{E}_{X|s}\left[ D_{\mathrm{KL}}\left( P(Y \mid X) \parallel \tilde{P}_s(Y \mid q(X), s) \right) \right]$$

$$= \mathbb{E}_{X|s}\left[ D_{\mathrm{KL}}\left( P(Y \mid X) \parallel \tilde{P}_s^*(Y \mid q_s^*(X), s) \right) \right]$$

$$= \mathbb{E}_{X|s}\left[ D_{\mathrm{KL}}\left( P(Y \mid X) \parallel \hat{P}(Y \mid \tilde{q}(X), s) \right) \right] + \mathbb{E}_{X|s}\left[ \sum_Y P(Y \mid X) \log \frac{\hat{P}(Y \mid \tilde{q}(X), s)}{\tilde{P}_s^*(Y \mid q_s^*(X), s)} \right]$$

$$= \mathbb{E}_{X|s}\left[ D_{\mathrm{KL}}\left( P(Y \mid X) \parallel \hat{P}(Y \mid \tilde{q}(X), s) \right) \right] - \mathbb{E}_{X|s}\left[ \sum_Y P(Y \mid X) \log \frac{\tilde{P}_s^*(Y \mid q_s^*(X), s)}{\hat{P}(Y \mid \tilde{q}(X), s)} \right]$$

$$= \mathbb{E}_{X|s}\left[ D_{\mathrm{KL}}\left( P(Y \mid X) \parallel \hat{P}(Y \mid \tilde{q}(X), s) \right) - D_{\mathrm{KL}}\left( \tilde{P}_s^*(Y \mid q_s^*(X), s) \parallel \hat{P}(Y \mid \tilde{q}(X), s) \right) \right]$$

$$\leq \mathbb{E}_{X|s}\left[ D_{\mathrm{KL}}\left( P(Y \mid X) \parallel \hat{P}(Y \mid \tilde{q}(X), s) \right) \right]$$

$$(9)$$

In the first equality we used the definition of $(\tilde{P}_s^*, q_s^*)$ as the solution to the minimization problem in the first equality. In the second equality, $\tilde{q} = g(s)$ for any querier $g$ and $\hat{P}(Y \mid \tilde{q}(X), s) = f(\{\tilde{q}, \tilde{q}(X)\} \cup s)$ for any classifier $f$. In the fourth equality we appealed to lemma 1 to conclude that $\tilde{P}_s^*(Y \mid q_s^*(X), s) = P(Y \mid q_s^*(X), s)$, the true posterior over $Y$ given answer $q_s^*(X)$ and history $s$. The final step we used the non-negativity of the KL-divergence for getting rid of the second term.

Since the inequality in equation 9 holds $\forall S = s$, and mappings $f$ and $g$, we conclude that $q_s^* = g^*(s)$ and $\tilde{P}_s^* = f^*(\{q_s^*, q_s^*(X)\} \cup s)$ for any given $S = s$. Equation 4 in the proposition is then proved by using lemma 1 to characterize $q_s^*$ and $\tilde{P}_s^*$.

## B  TRAINING PROCEDURE

Consider a mini-batch of $N$ samples, $\{(x_i, y_i)\}_{i=1}^N$ from a training set.

In Deep V-IP objective, the KL-divergence is mathematically equivalent to the cross entropy loss. The mini-batch estimate of this loss can be expressed as:

$$\min_{\theta, \eta} \quad -\frac{1}{N} \sum_{i=1}^N y_i \log \hat{y}_i$$
$$\text{subject to} \quad q_\eta = \texttt{argmax}(g_\eta(s_i))$$
$$\hat{y}_i = f_\theta(s_i \cup \{q_\eta, q_\eta(x_i)\}),$$
$$(10)$$

where $y_i$ is the ground truth label corresponding to input $x_i$. $s_i$ is obtained by sampling for $P_S^J$ as defined in §3.3. We optimize the above objective using Stochastic Gradient Descent (or its variants).

To optimize objective 10, for every sample $x_i$ in the batch, the sampled history $s_i$ is fed into the querier network $g_\eta$ which outputs a score for every query $q \in Q$. The `argmax(.)` (see D regarding its differentiability) operator converges these scores into a $|Q|$-dimensional one-hot vector, with the non-zero entry at the location of the max. We then append this argmax query $q_\eta$ and it's answer, $(q_\eta, q_\eta(X))$ to $s_i$. The updated history $s_i \cup (q_\eta, q_\eta(x_i))$ is then fed into the classifier $f_\theta$ to obtain a softmax distribution over the labels, denoted as $\hat{y}_i$.

## C  EXPERIMENT DETAILS

All of our experiments are implemented in python using PyTorch (Paszke et al., 2019) version 1.12. Moreover, all training is done on one computing node with 64-core 2.10GHz Intel(R) Xeon(R) Gold 6130 CPU, 8 NVIDIA GeForce RTX 2080 GPUs (each with 10GB memory) and 377GB of RAM.

**General Optimization Scheme.** The following optimization scheme is used in all experiments for both Initial Random Sampling and Subsequent Biased Sampling, unless stated otherwise. We minimize the Deep V-IP objective using Adam (Kingma & Ba, 2014) as our optimizer, with learning rate `lr=1e-4`, `betas=(0.9, 0.999)`, `weight_decay=0` and `amdgrad=True` (Reddi et al., 2019). We also use Cosine Annealing learning rate scheduler (Loshchilov & Hutter, 2016) with `T_max=50`. We train our networks $f_\theta$ and $g_\eta$ for 500 epochs using batch size 128. In both sampling stages, we linearly anneal temperature parameter $\tau$, in our straight-through softmax estimator, from 1.0 to 0.2 over the 500 epochs.

**Training Details for RAM and RAM+.** For train the RL policy and classification network we use the popular PPO algorithm (Schulman et al., 2017) with entropy regularization (0.01 regularization parameter). We used in initial learning rate of `3e-5` and clip value `0.2`. For a fair comparison the architectures for the policy[8] and classification networks are kept the same for RAM, RAM+ and V-IP.

### C.1  SPECIES CLASSIFICATION ON CUB.

**Dataset and Query Set.** Caltech-UCSD Birds-200-201 (CUB-200) (Wah et al., 2011) is a dataset of 200 bird species, containing 11,788 images with 312 annotated binary features for different visual attributes of birds, such as the color or shape of the wing or the head of the bird. We construct the query set $Q$ using these attributes. For example, given an image for a Blue-jay a possible query might be "is the back-colour blue?" and the answer "Yes".

The query set construction and data preprocessing steps as same as Chattopadhyay et al. (2022): In the original dataset, each annotated attribution for each image is too noisy, containing often imprecise description of the bird. Hence, if a certain attribute is true/false for over 50% of the samples in a given category, then that attribute is true/false for all samples in the same category. We also train a CNN to answer each query using the training set annotations called the concept network. This concept network provides answers for training all the methods compared in §4, namely, RAM, RAM+, G-IP and V-IP. Last but not least, our query set $Q$ consists of 312 queries that ask whether each of the binary attributes is 1 for present or $-1$ for absent.

**Architecture and Training** A diagram of the architectures is shown in Figure 6. Both $f_\theta$ and $g_\eta$ have the same full-connected network architecture (except the last linear layer), but they do not share any parameters with each. We initialize each architecture randomly, and train using the optimization scheme mentioned in the beginning of this section. The input history is a $|Q|$-dimensional vector with unobserved answers masked out with zeros.

**Updating the History.** Let the history of query-answer pairs observed after $k$ steps be denoted as $S_k$. $S_k$ is represented by a masked vector of dimension 312, and $q_{k+1} = \texttt{argmax}(g_\eta(S_k))$[9] is a one-hot vector of the same dimension, denoting the next query. For a given observation $x^{\text{obs}}$, we update $S_k$ using $q_{k+1}$ as follows:

- We obtain the query-answer by performing a point-wise multiplication, that is, $q_{k+1}(x^{\text{obs}}) = q_{k+1} \odot x^{\text{obs}}$.

- We update the history to $S_{k+1}$ by adding this query answer $q_{k+1}(x^{\text{obs}})$ to $S_k$.

The entire process can be denoted by $S_{k+1} = S_k + q_{k+1} \odot x^{\text{obs}}$.

---

[8]This term is from the RL community. In our context, the policy network is exactly the querier network.

[9]recall $g_\eta$ is our querier function parameterized by a deep network with weights $\eta$.

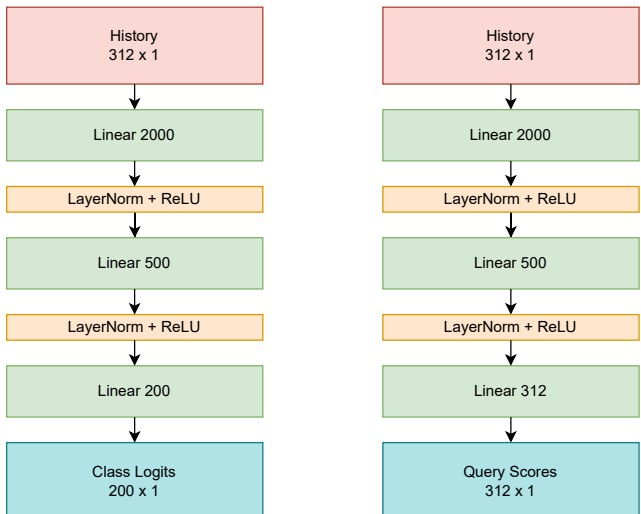

Figure 6: Classifier $f_\theta$ (Left) and querier $g_\eta$ (Right) architectures used for species classification on CUB.

## C.2  TOPIC IDENTIFICATION ON THE HUFFINGTON POST NEWS CATEGORY DATASET.

**Dataset and Query Set.** The Huffington Post News Category dataset (HuffingtonNews) is a natural language dataset, containing news "headlines" and their short descriptions (continuation of the headline) extracted from Huffington Post news published between 2012 and 2018. We follow the same data-processing procedure as Chattopadhyay et al. (2022): Each data point is an extended headline formed by concatenation of the headline with their short descriptions. Moreover, we also remove redundant categories, including semantically ambiguous and HuffPost-specific words such as "Impact" and "Worldpost." We also remove categories with small number of articles, along with semantically equivalent category names, such as "Arts & Culture" versus "Culture & Art." After processing, there is a total of 10 categories in the dataset. In addition, only the top-1,000 words are kept according to their tf-idf scores (Lavin, 2019), along with semantically redundant words removed. For more details, please refer to Chattopadhyay et al. (2022).

The query set $Q$ contains binary questions of whether one of the 1000 words exist in the headline. The query answer is $1$ if the word in question is present and $-1$ if absent.

**Architecture and Training.** A diagram of the architectures is shown in Figure 7. Both the classifer $f_\theta$ and the querier $g_\eta$ shares the same architecture except the last layer; However, they do not share any parameters. The inputs to $f_\theta$ and $g_\eta$ are masked vectors, with masked values set to 0. To optimize the Deep V-IP objective, we randomly randomly initialize $f_\theta$ and $g_\eta$, and train using Adam optimizer and Cosine Annealing learning rate scheduler, with the settings mentioned in the beginning of the section. During Subsequent Adaptive Sampling, we train for 100 epochs using Stochastic Gradient Descent (SGD) instead of Adam, with learning rate `lr=1e-4` and `momentum=0.9`, and Cosine Annealing learning rate scheduler, with `T_max=100`. The input history is a $|Q|$-dimensional vector with unobserved answers masked out with zeros.

**Updating the History.** The method of updating the history is equivalent to that for CUB as mentioned in §C.1. The history is now a masked vector of dimension 1000, since there are 1000 queries in our query set for this dataset.

## C.3  IMAGE CLASSIFICATION ON MNIST, KMNIST AND FASHION-MNIST.

Each setting mentioned in the following is the same for MNIST, KMNIST and Fashing-MNIST unless stated otherwise.

**Dataset and Query Set.** MNIST (LeCun et al., 1998), KMNIST (Clanuwat et al., 2018), Fashion-MNIST (Xiao et al., 2017b) are gray-scale hand-written digit datasets, each containing 60,000 train-

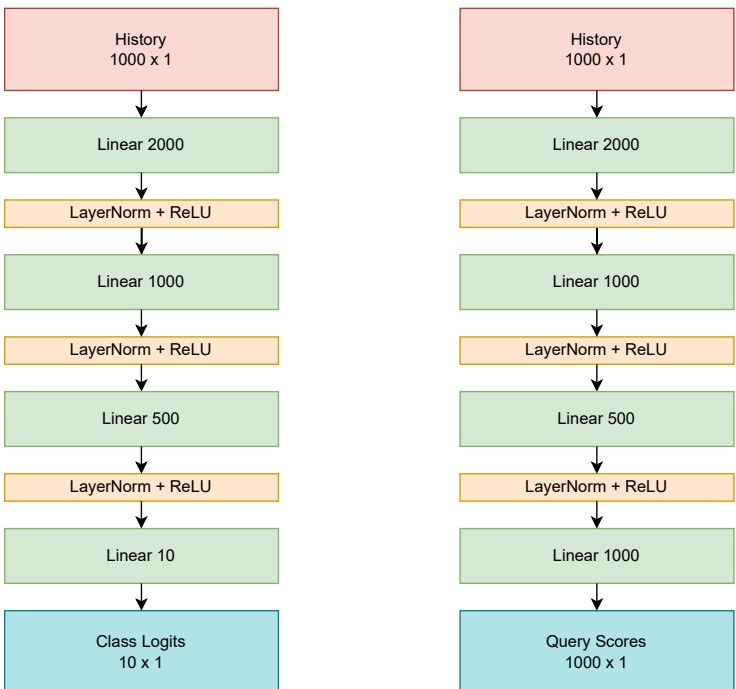

Figure 7: Classifier $f_\theta$ (Left) and querier $g_\eta$ (Right) architectures used for topic identification on Huffington Post News Category dataset.

ing images and 10,000 testing images of size $28 \times 28$. We follow Chattopadhyay et al. (2022) for data pre-processing procedure and the design of our query sets of all three datasets.

Each gray-scale image is converted into a binary image. For MNIST and KMNIST, we round values greater or equal than 0.5 up to 1 and round values below 0.5 down to -1. For Fashion-MNIST, we round values greater or equal up to 0.1 to 1 and round values below 0.1 down to -1.

Each query set $Q$ contains all overlapping $3 \times 3$ patches over the $28 \times 28$ pixel space, resulting in 676 queries. Each query answer indicates the 9 pixel intensities at the queried patch. The inputs to the classifier $f_\theta$ and the querier $g_\theta$ are masked images, with masked pixels zeroed out if they are not part of the current History.

**Updating the History.** The method for updating the history is equivalent to that for CUB as mentioned in §C.1 with some differences as we will describe next. For a given observation $x^{\text{obs}}$, we update $S_k$ using $q_{k+1}$ as follows:

- We reshape $q_{k+1}$ from a vector of dimension 676 (the number of queries) to a 2D grid of dimension $26 \times 26$, denoted by $\hat{q}_{k+1}$.
- $\hat{q}_{k+1}$ is then converted to a binary matrix with 1s at the location corresponding to the queried $3 \times 3$ patch and 0s everywhere else via a convolutation operation with a kernel of all 1s of size $3 \times 3$, stride 1 and padding 2.
- We then obtain the query-answer by performing a hadamard product of the convolved output (in the previous step) with $x^{\text{obs}}$. This results in a masked image, $\hat{q}_{k+1}(x^{\text{obs}})$, with the queried patch revealed and 0 everywhere else.
- Finally, we update the history to $S_{k+1}$ by adding this query-answer to $S_k$. To account for pixels observed (unmsaked) in $\hat{q}_{k+1}(x^{\text{obs}})$ that overlap with the history $S_k$, we clip the values in $S_k$ to lie between $-1$ and $1$.

The entire process can be summarized as,

$$S_{k+1} = \texttt{Clip}\left(S_k + (\texttt{Conv2D}(\hat{q}_{k+1}) \odot x^{\text{obs}}), \ \texttt{minval} = -1, \ \texttt{maxval} = 1\right)$$

.

**Architecture and Training.** Refer to Figure 8 for a diagram of the architecture for the classifier $f_\theta$ and the querier $g_\eta$. Every `Conv` is a 2D Convolution with a $3 \times 3$ kernel, stride 1 and padding 1. Moreover, every `MaxPool` is a 2D max pooling operator with a $2 \times 2$ kernel. We initialize each architecture from random initialization, and train our networks using Adam as our optimizer and Cosine Annealing learning rate scheduler, with the same settings mentioned in the beginning of this section.

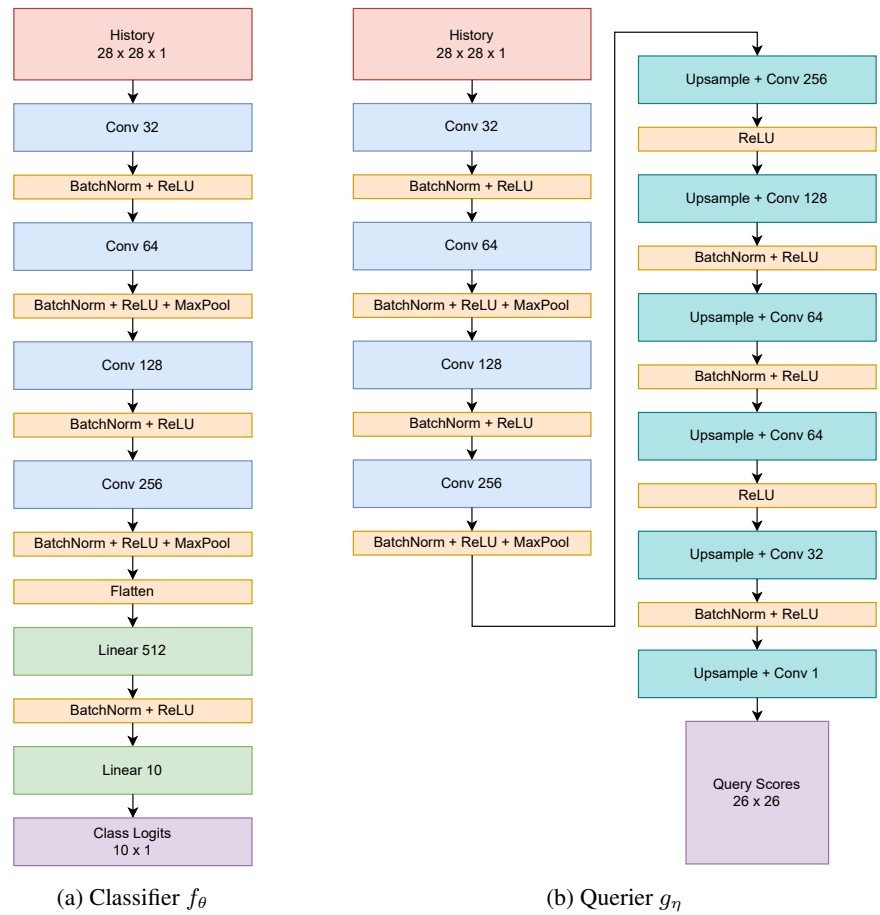

(a) Classifier $f_\theta$      (b) Querier $g_\eta$

Figure 8: Architectures used for image classification on MNIST, KMNIST, and Fashion-MNIST.

## C.4   MEDICAL DIAGNOSIS ON SYMCAT, MUZHI AND DXY

**Dataset and Query Set.** MuZhi (Wei et al., 2018) and Dxy (Xu et al., 2019) are two real-world medical datasets containing symptoms and diseases extracted from Chinese healthcare websites (`https://muzhi.baidu.com/` and `https://dxy.com/`), where doctors provide online help based on patient's self-report symptoms and online conversations. We follow the same data processing procedure as He et al. (2022). MuZhi has 66 symptoms (features) and 4 diseases (classes), including children's bronchitis, children's functional dyspepsia, infantile diarrhea infection, and upper respiratory infection. Dxy dataset contains 41 symptoms for 5 diseases: allergic rhinitis, upper respiratory infection, pneumonia, children hand-foot-mouth disease, and pediatric diarrhea. Last but not least, our query set $Q$ consists of queries that correspond to asking questions about the presence of each symptom for the patient; the query-answer is either 1 for Yes, 0 for No and -1 for Can't Say.

SymCAT is a synthetic medical dataset generated from a symptom checking website called SymCAT introduced by Peng et al. (2018). The dataset has three versions; SymCAT-200 contains 328 symptoms and 200 disease, SymCAT-300 contains 349 symptoms and 300 classes, and SymCAT-400 con-

tains 355 symptoms and 400 classes. We used publically aviailable version of this dataset provided by Nesterov et al. (2022) at `https://github.com/SympCheck/NeuralSymptomChecker`. Our query set $Q$ consists of queries that correspond to asking questions about the presence/absence of each symptom for the patient; the query-answer is either 1 for Yes and 0 for No.

**Architecture and Training.** A diagram of the architecture is shown in Figure 9. We used the set architecture proposed in Ma et al. (2018). We made this choice for a fair comparison with He et al. (2022) which also used to same architecture for their partial-VAEs. The input to the network is a concatenation of the query-answers $\{$`q(x_j) : q` $\in Q\}$, trainable positional embeddings `e_j` (red blocks), and bias terms `b_j` (blue blocks). Each positional embedding is also multiplied by the query-answer. After the first linear layer, the intermediate embedding is multiplied by a query-answer mask derived from the history, which each dimension has a value of 1 for query selected in the history and 0 otherwise. To optimize our objective, we randomly initialize $f_\theta$ and $g_\theta$ and train using the same optimizer and learning rate scheduler setting as mentioned in the beginning of the section. However, we train our algorithms for only 200 epochs, and linearly anneal the straight-through softmax estimator's temperature $\tau$ from 1.0 to 0.2 over the first 50 epochs.

**Updating the History.** Let the history of query-answer pairs observed after $k$ steps be denoted as $S_k$. Since we used set architectures as our querier and classifier networks for these datasets, as proposed in Ma et al. (2018), $S_k$ is represented as a set consisting of embeddings of query-answer pairs observed so far. The next query, $q_{k+1} = $ `argmax`$(g_\eta(S_k))$ is a one-hot vector of dimension equal to the size of the query set used. For a given observation $x^{\text{obs}}$, we update $S_k$ using $q_{k+1}$ as follows:

- Let $M$ be a matrix of size $|Q| \times d$ where every row corresponds to a query-answer evaluated at $x^{\text{obs}}$ and $d$ is the size the embeddings used for representing the query-answers. We obtain the answer corresponding to the selected query $q_{k+1}$ by performing a matrix-vector product, that is, $q_{k+1}(x^{\text{obs}}) = q_{k+1}^T M$.

- We update the history to $S_{k+1}$ by concatenating $q_{k+1}(x^{\text{obs}})$ to $S_k$.

## C.5 IMAGE CLASSIFICATION ON CIFAR-10 AND CIFAR-100

**Dataset and Query Set.** CIFAR-$\{10,100\}$ (Krizhevsky et al., 2009) are natural image datasets that contain $\{10, 100\}$ different classes of objects. They contain 50,000 training images and 10,000 testing images. Each RGB image is of size $32 \times 32$. We design our query set $Q$ following Rangrej & Clark (2021) consisting of all $8 \times 8$ overlapping patches with stride 4. This results in a query set size $|Q|$ of 49. The inputs to the classifier $f_\theta$ and the querier $g_\eta$ are full-sized $3 \times 32 \times 32$ masked image, with masked pixels zeroed out if they are not part of the current History.

**Architecture and Training.** For CIFAR-10, the architectures used for the classifier $f_\theta$ and querier $g_\eta$ are both Deep Layer Aggregation Network (DLA) (Yu et al., 2018). $f_\theta$ and $g_\eta$ do not share any parameters with each other. An out-of-the-box implementation was used and can be found here: `https://github.com/kuangliu/pytorch-cifar/blob/master/models/dla.py`. For CIFAR-100, the architectures used for the classifier $f_\theta$ and querier $g_\eta$ are both DenseNet169 (Huang et al., 2017). $f_\theta$ and $g_\eta$ do not share any parameters with each other. An out-of-the-box implementation was used and can be found here: `https://github.com/kuangliu/pytorch-cifar/blob/master/models/densenet.py`. The only change made to the architectures is the last layer, which the dimensions depend on the number of classes or the size of the query set $Q$.

During training, we follow standard data processing techniques as in He et al. (2016). In CIFAR-10, we set batch size 128 for both initial and subsequent sampling stages. In CIFAR-100, we set batch size 64 for both initial and subsequent sampling stages. For both CIFAR-10 and CIFAR-100, we randomly initialize $f_\theta$ and $g_\eta$, and train them for 500 epochs during Initial Random Sampling using optimizer Adam and Cosine Annealing learning rate scheduler, with settings mentioned above. During Subsequent Adaptive Sampling, we optimize using Stochastic Gradient Descent (SGD), with learning rate `lr=0.01` and `momentum=0.9` and Cosine Annealing learning rate scheduler, with `T_max=50`, for 100 epochs.

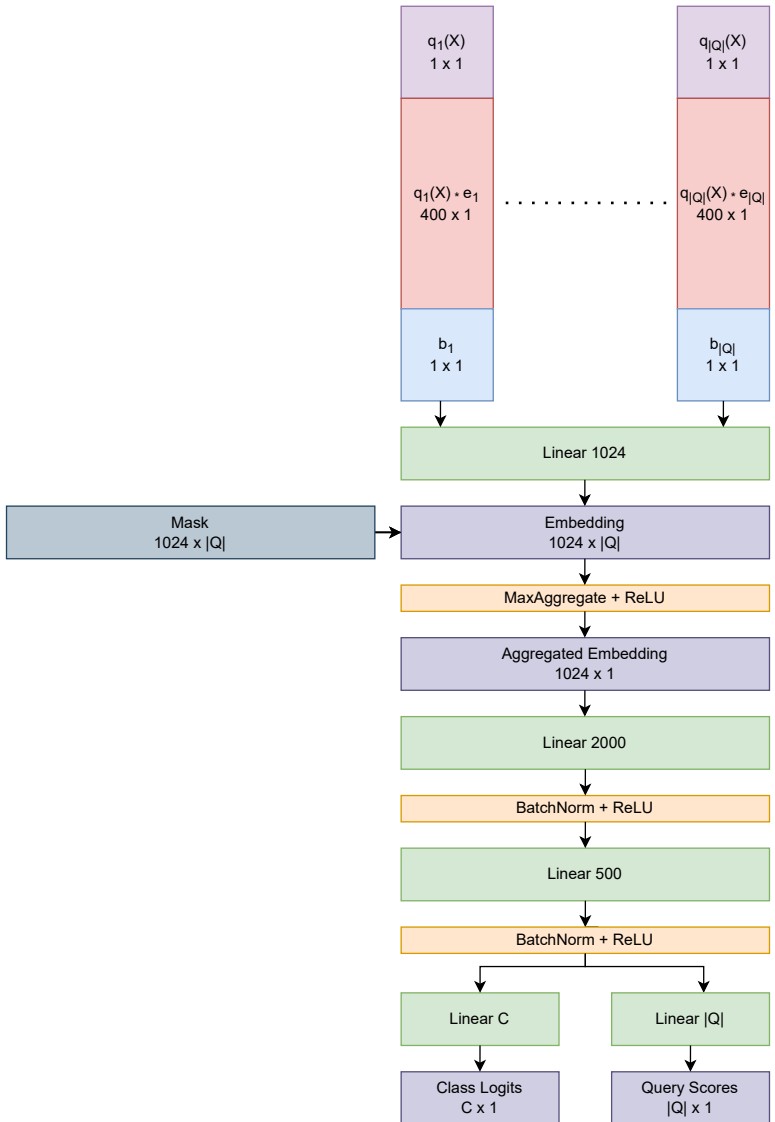

Figure 9: Architecture used for Medical Diagnosis tasks on SymCAT, MuZhi and Dxy. The classifier $f_\theta$ and querier $g_\eta$ share parameters in this architecture. $|Q|$ is the size of the query set, which also corresponds to the number of features. $C$ is the number of classes, depending on the dataset.

**Updating the History.** The method of updating the history is similar to that for MNIST, KMNIST, and Fashion-MNIST, as mentioned in § C.3. For a given observation $x^{\text{obs}}$, we update the history $S_k$ using $q_{k+1}$ as follows:

- We reshape $q_{k+1}$ from a vector of dimension 49 (the number of queries) to a 2D grid of dimension $7 \times 7$, denoted by $\hat{q}_{k+1}$.

- $\hat{q}_{k+1}$ is then converted to a binary matrix with 1s at the location corresponding to the queried $8 \times 8$ patch and 0s everywhere else via a 2D transposed convolutation operation with a kernel of all 1s of size $8 \times 8$, stride 4 and no padding.

- We then obtain the query-answer by performing a hadamard product of the convolved output (in the previous step) with $x^{\text{obs}}$. This results in a masked image, $\hat{q}_{k+1}(x^{\text{obs}})$, with the queried patch revealed and 0 everywhere else.

- Finally, we update the history to $S_{k+1}$ by adding this query-answer to $S_k$. Any pixel $ij$ that is observed (unmasked) in $\hat{q}_{k+1}(x^{\text{obs}})$ and is also observed in the history $S_k$ is handled by the transformation $S_{k+1}[i,j] \rightarrow S_k[i,j]$ if $S_{k+1}[i,j] = 2S_k[i,j]$.

The entire process can be summarized as,

$$S'_{k+1} = S_k + \texttt{TransposedConv2D}(\hat{q}_{k+1}) \odot x^{\text{obs}}.$$

$\forall (i,j) \in \{\text{Number of pixels in image}\}$

$$\begin{aligned} S_{k+1}[i,j] &= S_k[i,j] &&\text{if } S'_{k+1}[i,j] = 2S_k[i,j] \quad (11)\\ S_{k+1}[i,j] &= S'_{k+1}[i,j] &&\text{otherwise} \end{aligned}$$

## D  STRAIGHT-THROUGH SOFTMAX ESTIMATOR

As mentioned in §3.3, in the main text, we employ the straight-through softmax gradient estimator for differentiating through the $\texttt{argmax}$ operation. We will now describe this estimator in detail. Consider the following optimization problem,

$$\min_{\theta \in \mathbb{R}^d} f(\texttt{argmax}(\theta)) \tag{12}$$

Let $Z := \texttt{argmax}(\theta)$. We will assume $f$ is differentiable in its input. The straight-through softmax estimator of the gradient of $f$ w.r.t $\theta$ is defined as,

$$\nabla_\theta^{ST} f := \frac{\partial f}{\partial Z} \frac{d\text{softmax}_\tau(\theta)}{d\theta},$$

where $\text{softmax}_\tau(\theta) := \left[ \frac{e^{\frac{\theta_1}{\tau}}}{\sum_{i=1}^d e^{\frac{\theta_i}{\tau}}} \quad \frac{e^{\frac{\theta_2}{\tau}}}{\sum_{i=1}^d e^{\frac{\theta_i}{\tau}}} \quad \cdots \quad \frac{e^{\frac{\theta_d}{\tau}}}{\sum_{i=1}^d e^{\frac{\theta_i}{\tau}}} \right]$ and $\tau$ is the temperature parameter. Notice that $\lim_{\tau \to 0} \text{softmax}_\tau(\theta) \to \texttt{argmax}(\theta)$. Thus, we replace the gradient of the argmax operation which is either 0 almost everywhere or doesn't exist with a surrogate biased estimate.

Equation 12 can then be optimized using the straight-through estimator by iteratively carrying out the following operation,

$$\theta = \theta - \eta \nabla_\theta^{ST} f,$$

where $\eta$ is the learning rate.

In our experiments we start with $\tau = 1.0$ and linearly anneal it down to 0.2 over the course of training.

## E  ABLATION STUDIES

In §3.3 we discussed two possible architectures for operating on histories of arbitrary sequence lengths; the set-based architectures (as in Figure 9) and the fixed-sized input masking based architectures (as in Figure 6). In Figure 10, we compare the two architectures on the same task of bird species identification using the same query set (see Table 1). We see that the fixed-sized input masking based architecture performs better in terms of avg. number of queries needed to get the same performance. Based on this observation, we use the latter architecture in all our experiments except on the medical datasets, where we used the set-based architecture for fair comparison with the BSODA method which uses a similar set-based architecture for their partial-VAEs.

In §3.3 we discussed that a sampling distribution $P_S$ that assigns a positive mass to every element in $\mathbb{K}$ would make learning a good querier function challenging since the network has to learn over an exponentially (in size of Q) large number of histories. Instead we proposed to sequential bias the sampling according to our current estimate of the querier function. We validate the usefulness of this biased sampling strategy in Figure 11. In most datasets we observe that biasing the sampling distribution for $S$ helps learn better querying strategies (in terms of accuracy vs. explanation length trade-offs).

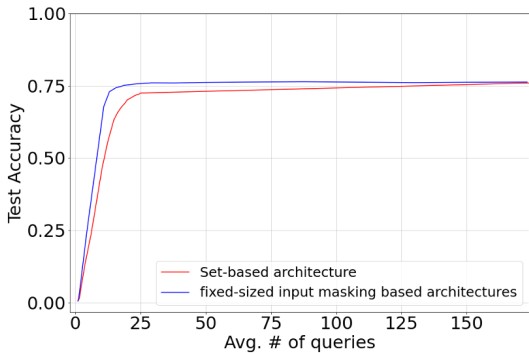

Figure 10: Comparison of accuracy v/s explanation length (avg. number of queries) curves for the set-based and fixed-sized input masking based architectures on the CUB-200 dataset.

Notice that in most datasets, biased sampling without the initial random sampling ultimately ends up learning a slightly better strategy (in terms of accuracy vs. explanation length trade-offs). However, since biased sampling requires multiple forward passes through our querier network to generate histories, it is much slower than the initial random sampling (IRS) scheme for $P_S$. Thus, when computational resources are not a concern one could do away with the initial random sampling but under a computational budget, the random sampling allows for finding a quick solution which can then be finetuned using biased sampling. This finetuning will potentially require fewer epochs to get good performance than training using biased sampling from scratch.

## F  EXTENDED RESULTS

In Figure 12, we show the trade-off between accuracy and explanation length (avg. number of queries) on KMNIST and Fashion-MNIST datasets as the stopping criterion $\epsilon$ is changed. In both these datasets, the "MAP criterion" is used as the stopping criterion. V-IP performs better than RL-based RAM and RAM+ on both these datasets. V-IP is competitive with G-IP eventually surpassing it in terms of accuracy for longer query-answer chains.

In addition, Table 4 shows extended results for AUC values for test accuracy versus explanation length curves for different datasets. A simplified table is shown in Table 2.

Table 4: Extended results for Table 2. Every method, except G-IP, was repeated 5 times with a different seed. The high computational cost of G-IP (one run of inference on the MNIST test set takes a few weeks) makes it infeasible to repeat G-IP multiple times for computing standard-deviation values.

| Dataset | Random | RAM | RAM+ | G-IP | V-IP (Ours) |
|---|---|---|---|---|---|
| CUB | $0.557 \pm 0.002$ | $0.662 \pm 0.002$ | $0.695 \pm 0.004$ | **0.736** | $0.716 \pm 0.008$ |
| HuffingtonNews | $0.423 \pm 0.015$ | $0.389 \pm 0.010$ | $0.431 \pm 0.003$ | **0.691** | $0.664 \pm 0.002$ |
| MNIST | $0.868 \pm 0.002$ | $0.916 \pm 0.005$ | $0.920 \pm 0.007$ | **0.964** | $0.956 \pm 0.002$ |
| KMNIST | $0.775 \pm 0.003$ | $0.832 \pm 0.003$ | $0.841 \pm 0.002$ | 0.872 | **0.911** $\pm 0.008$ |
| Fashion-MNIST | $0.735 \pm 0.029$ | $0.770 \pm 0.003$ | $0.804 \pm 0.010$ | 0.831 | **0.849** $\pm 0.010$ |

## G  COMPARING V-IP WITH BLACK-BOX DEEP NETWORKS

An important aspect of the framework introduced by Chattopadhyay et al. (2022) is that the end-user defines queries that are interpretable to them. Given this set of queries, $Q$, V-IP learns to efficiently compose them into concise explanations for model predictions (in terms of query-answer chains). This begs the question, how much do we loose in terms of performance by constructing an interpretable $Q$? In Table 5, we report results studying this question.

"Acc. w/ V-IP given $\epsilon$" is the test accuracy obtained by V-IP after termination using our stopping criterion. "Acc. w/ V-IP given Q(X)" is the accuracy the classifier network (trained jointly with the

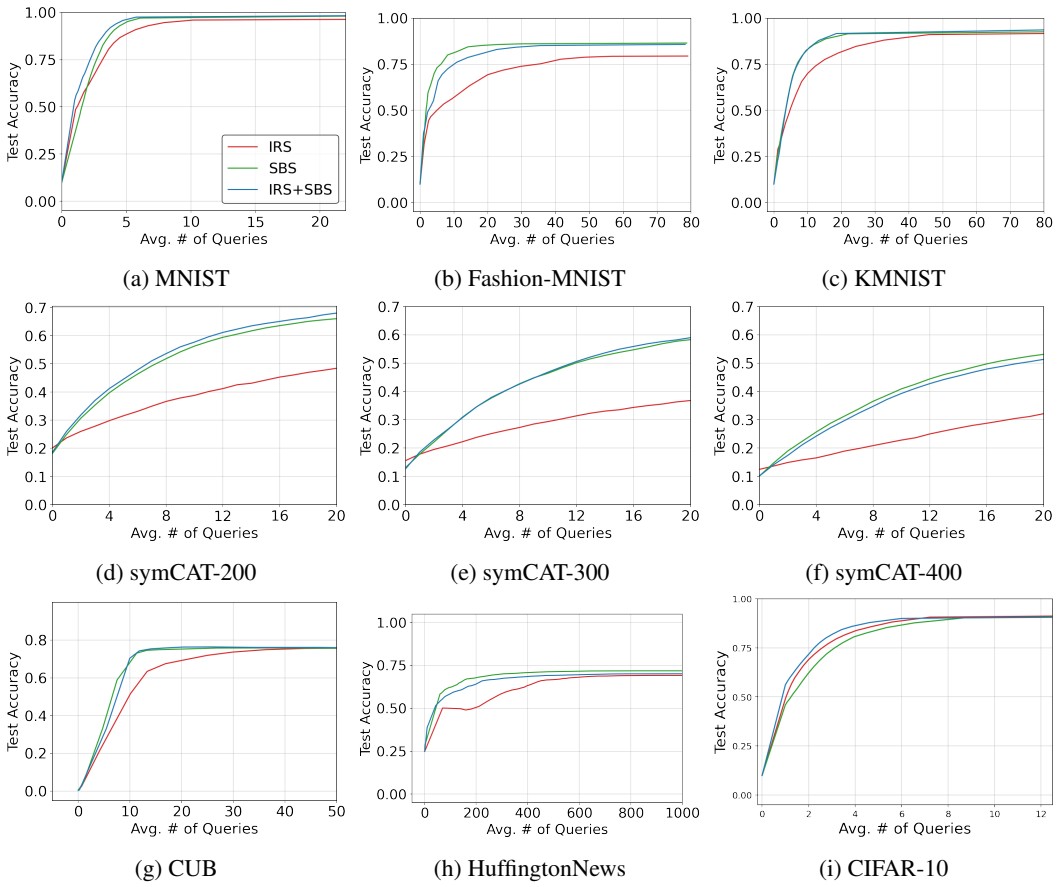

Figure 11: Accuracy v/s Explanation length (avg. number of queries) curves for different datasets with and without the successive biased sampling for history $S$. We denote training with Initial Random Sampling only as IRS; training with Subseqeuent Biased Sampling only as SBS; and training with IRS, followed by subsequent fine-tuning with SBS as IRS + SBS.

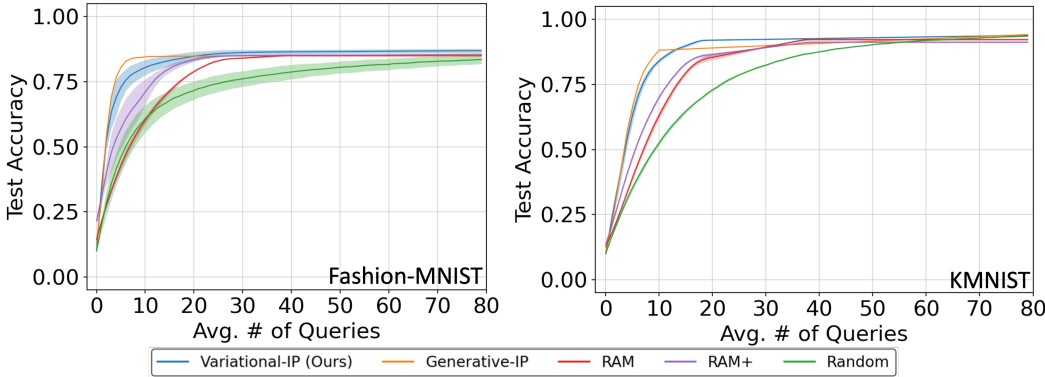

Figure 12: Tradeoff between accuracy and explanation length (average number of queries) for Fashion-MNIST (left) and KMNIST (right). Reported curves are averaged over 5 runs with shaded region denoting the standard-deviation.

queried network using the Deep V-IP objective) obtains when seeing all the query answers $Q(X)$. In all data sets, V-IP learns to predict with short explanations (avg. number of queries) and a test accuracy in proximity to what can be achieved if all answers were observed (col 6).

'Acc. w/ Black-Box given $Q(X)$" reports the accuracy a black-box deep network obtains by training on feature vectors comprised of all query answers $Q(X)$ using the standard cross-entropy loss. Columns 6 and 7 show that training a classifier network with the Deep V-IP objective results in only a minor loss in performance compared to training using the standard supervised classification cross-entropy loss.

"Acc. w/ Black-Box given $X$" reports test accuracies obtained by training black-box deep networks on the whole input $X$ to produce a single output (the classification label). Comparing these values with the accuracies reported in col 6 we see that basing predictions on an interpretable query set, almost always, results in a drop in accuracy. This is expected since interpretability can be seen as a constraint on learning. For example, there is a drop of about $15\%$ for the HuffingtonNews dataset since our queries are about the presence/absence of words which completely ignores the linguistic structure present in documents. Similarly, for the binary image classification tasks (MNIST, KM-NIST and Fashion-MNIST) the queries are binary patches which can be easily interpreted as edges, foregrounds and backgrounds. This binarization however results in a drop in performance, especially in Fashion-MNIST where it is harder to distinguish between some classes like coat and shirt without grayscale information.

Table 5: Comparison of Test Performance between prediction using V-IP with stopping criterion $\epsilon$, prediction using all queries, and prediction using a black-box trained model.

| Dataset | $\|Q\|$ | Stopping Criterion $(\epsilon)$ | Explanation Length | Acc. w/ V-IP given $\epsilon$ | Acc. w/ V-IP given $Q(X)$ | Acc. w/ Black-Box given $Q(X)$ | Acc. w/ Black-Box given $X$ |
|---|---|---|---|---|---|---|---|
| HuffingtonNews | 1000 | MAP (0.99) | 195.89 | 0.672 | 0.712 | 0.715 | 0.865 [10] |
| CUB-200 | 312 | Stability (0.001) | 18.82 | 0.752 | 0.763 | 0.763 | 0.827 [11] |
| MNIST | 676 | MAP (0.99) | 6.34 | 0.971 | 0.991 | 0.992 | 0.998 [12] |
| KMNIST | 676 | MAP (0.99) | 20.34 | 0.916 | 0.958 | 0.951 | 0.988 [13] |
| Fashion-MNIST | 676 | MAP (0.99) | 19.42 | 0.872 | 0.876 | 0.884 | 0.967 [14] |
| CIFAR-10 | 49 | Stability (0.127) | 9.79 | 0.936 | 0.946 | 0.955 | 0.955 [15] |
| CIFAR-100 | 49 | Stability (1.438) | 12.82 | 0.752 | 0.788 | 0.798 | 0.798 [16] |

# H    ADDITIONAL QUERY-ANSWER CHAINS

We show additional trajectories for different task and datasets: CUB-200 (Figure 13), MNIST (Figure 14), Fashion-MNIST (Figure 15), KMNIST (Figure 16), HuffingtonNews (Figure 18), CIFAR-10 (Figure 19) and CIFAR-100 (Figure 20). In every figure, we see that the correct predictions are explained by an interpretable sequence of query-answer pairs. The evolution of the posterior $P(Y \mid q_{1:k}(x^{\text{obs}}))$, as more and more queries are asked, gives insights into the model's decision-making process as we see it's belief shift among the possible labels for $x^{\text{obs}}$. This adds an additional layer of transparency to the model.

---

[10]Number taken from Chattopadhyay et al. (2022) where authors fine-tuned a Bert Large Uncased Transformer model to classify documents.

[11]Number taken from Koh et al. (2020) which trained a CNN on raw images of birds from the CUB-200 dataset

[12]Number reported from Hu et al. (2018)

[13]Number reported from Clanuwat et al. (2018)

[14]Number reported from Xiao et al. (2017a)

[15]Trained a Deep Layer Aggregation model Yu et al. (2018) to classify CIFAR-10 images from scratch.

[16]Number reported fromHuang et al. (2017)

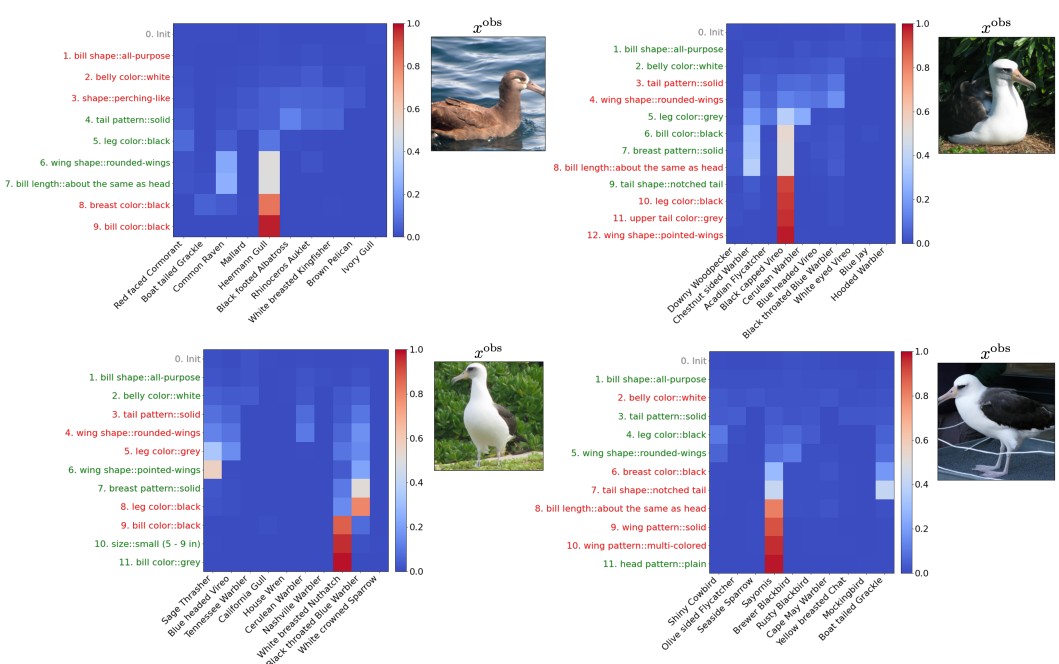

Figure 13: Examples for CUB-200.

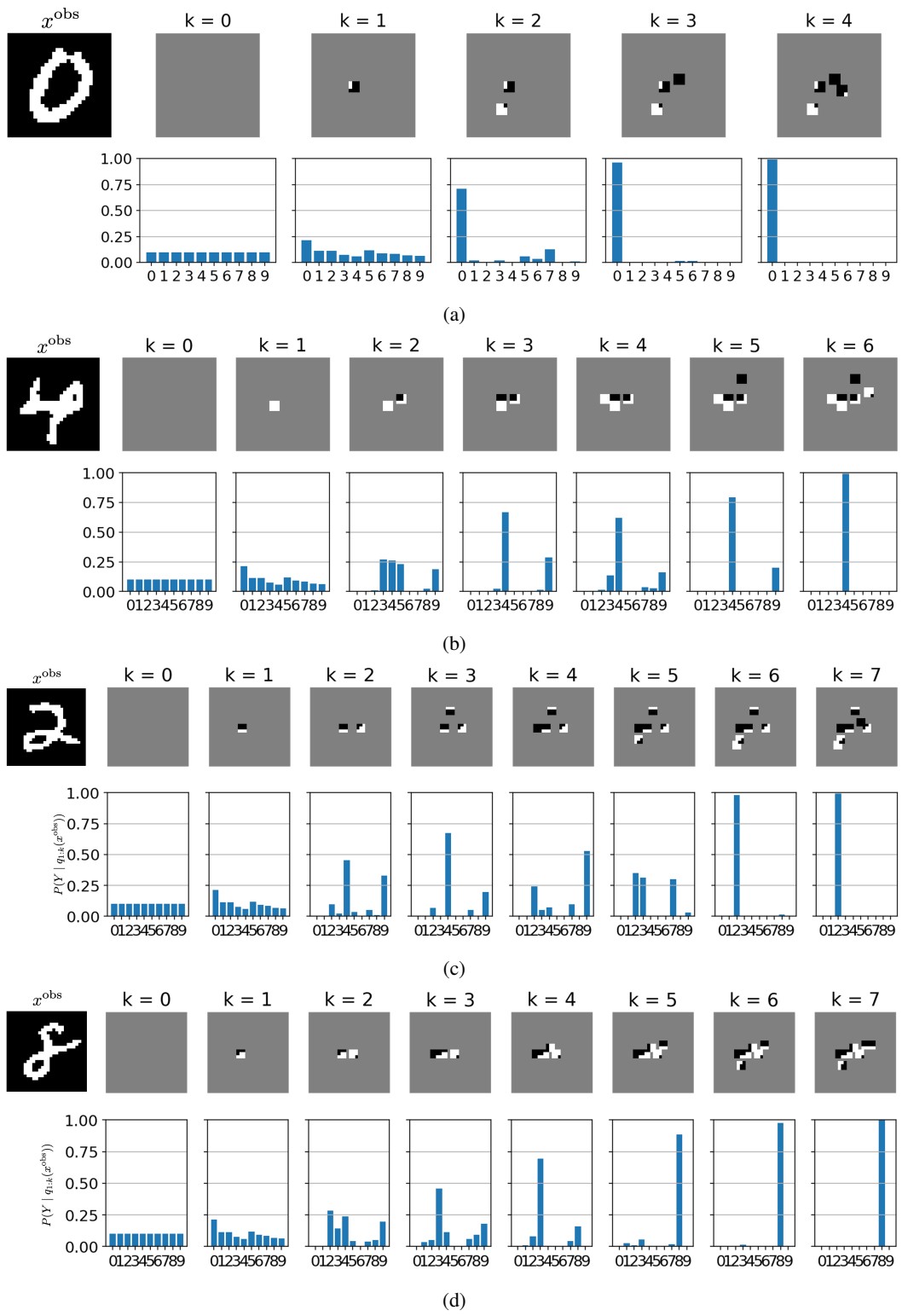

Figure 14: Examples for MNIST.

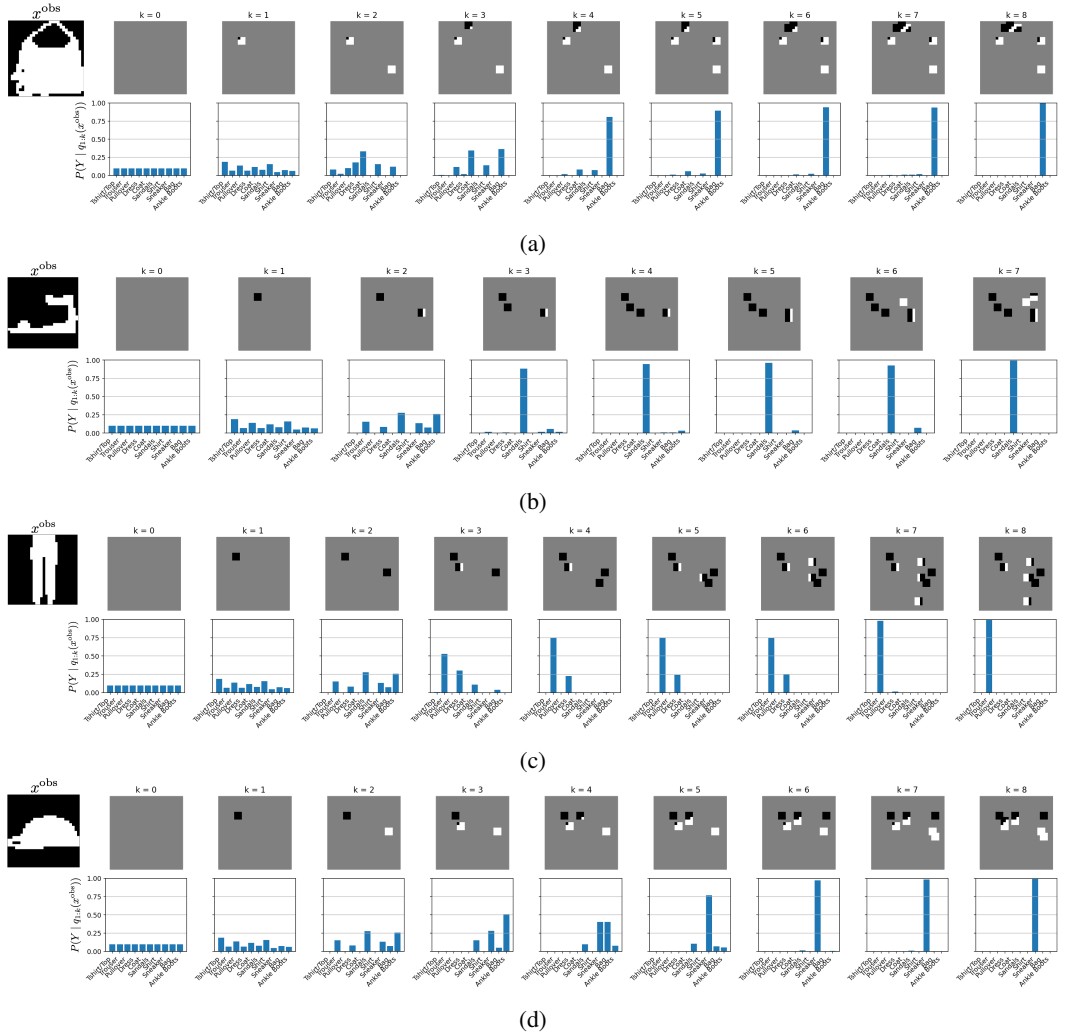

Figure 15: Examples for Fashion-MNIST.

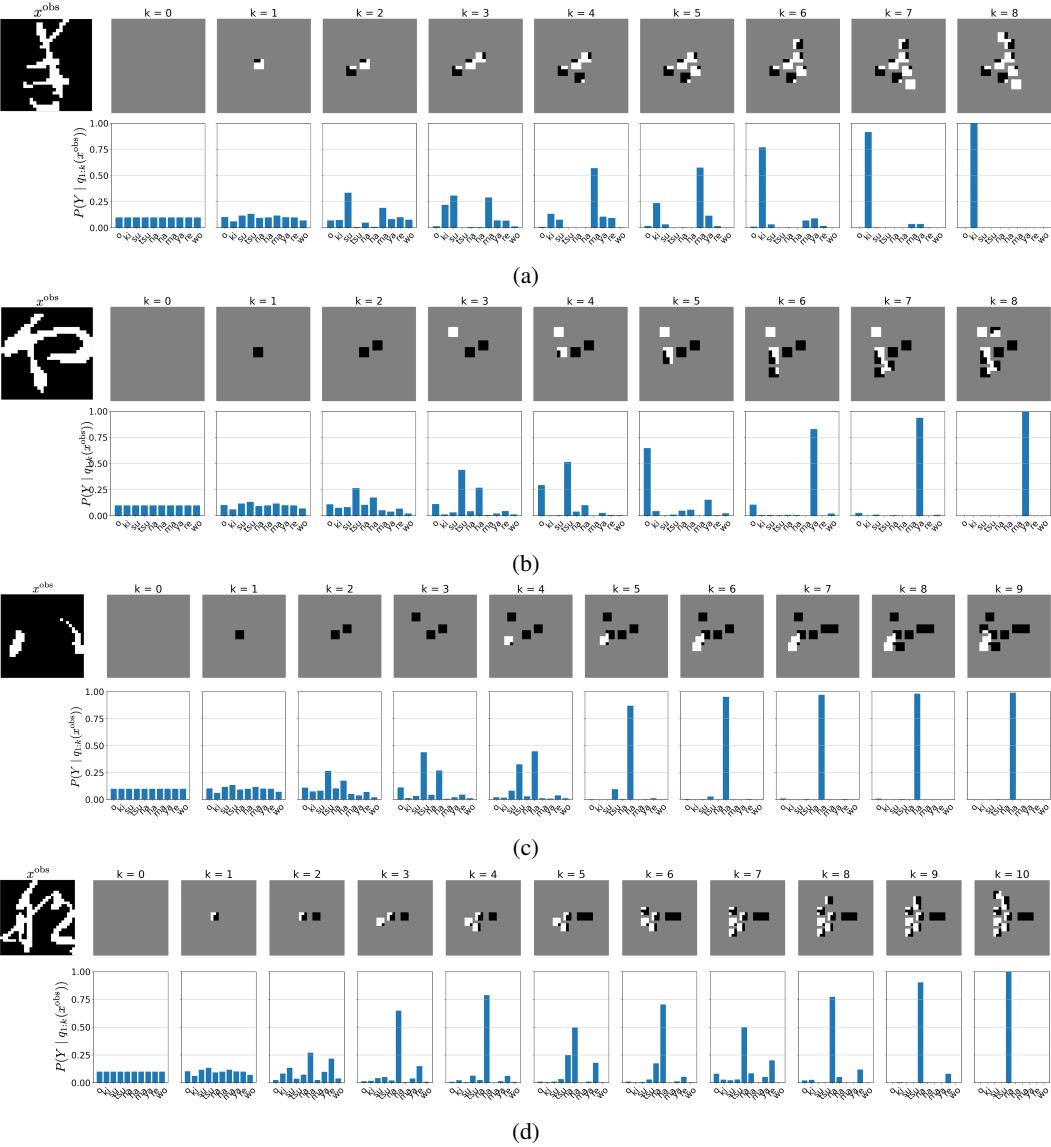

Figure 16: Examples for KMNIST.

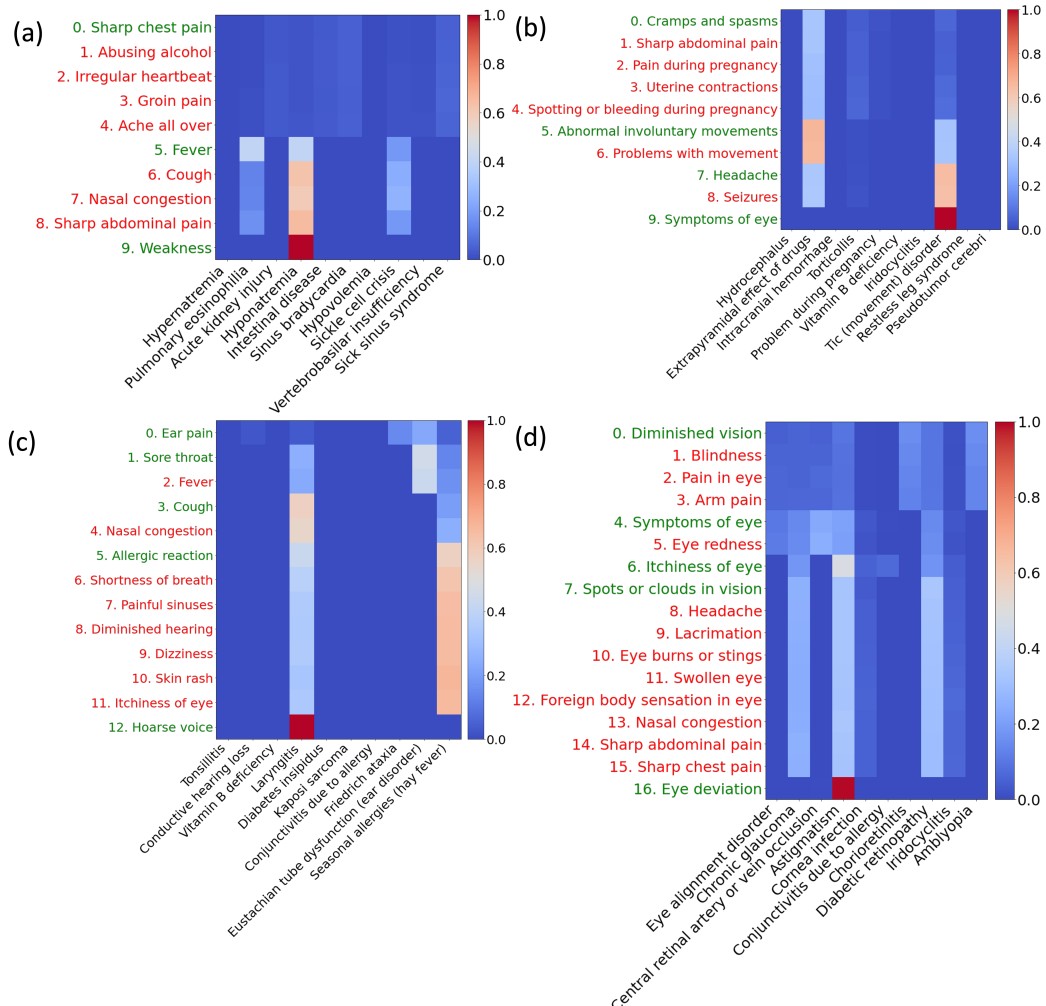

Figure 17: Examples for SymCAT200

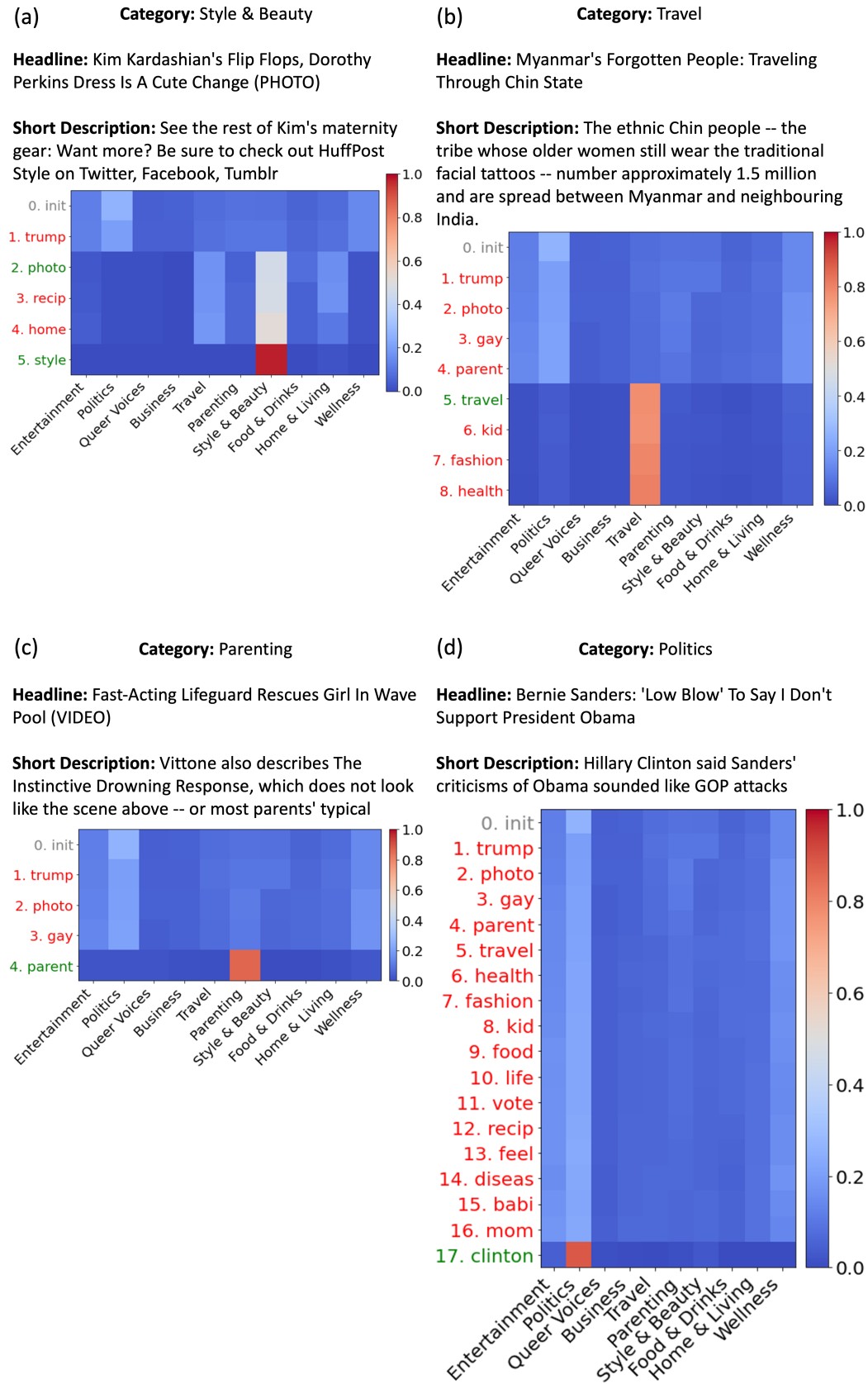

Figure 18: Examples for HuffingtonNews.

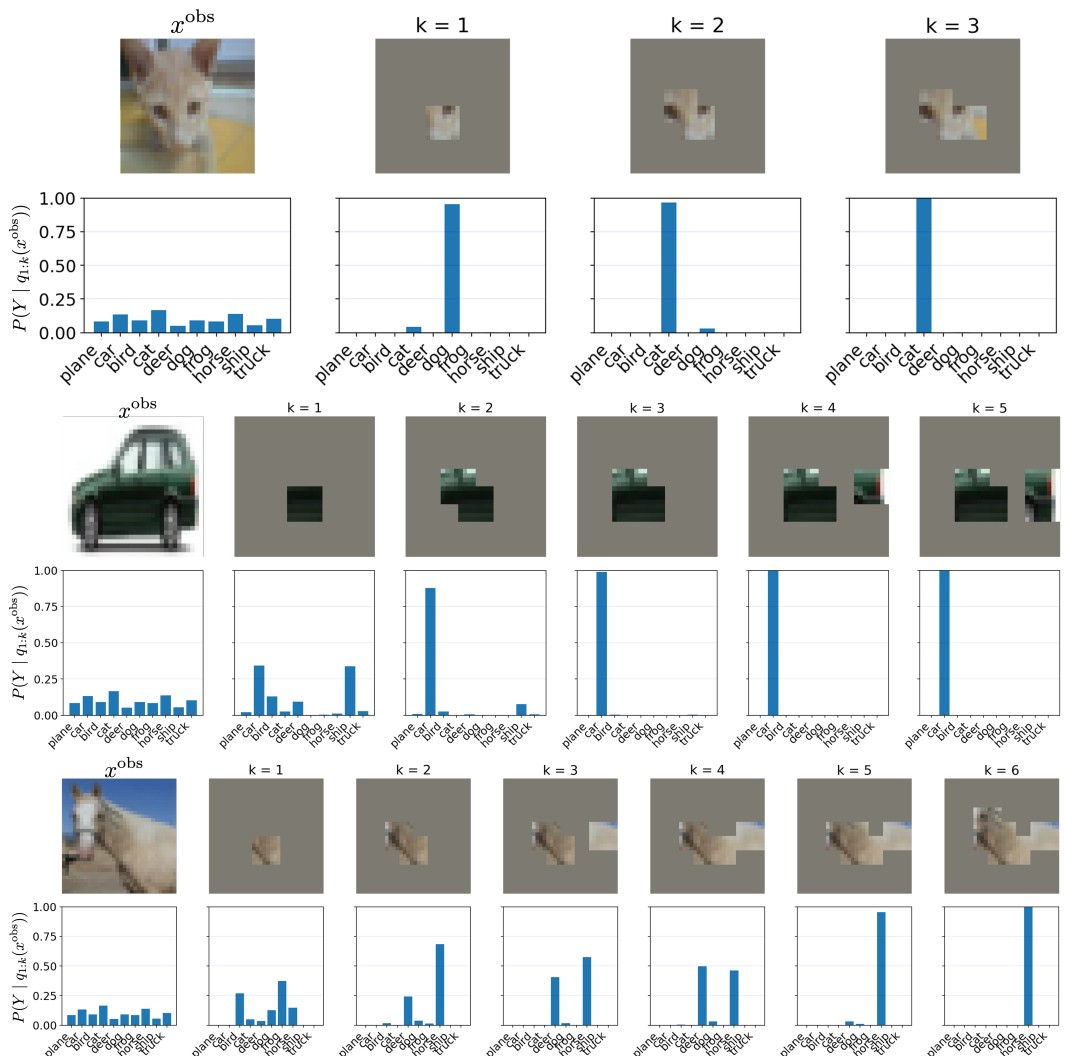

Figure 19: Examples for CIFAR10.

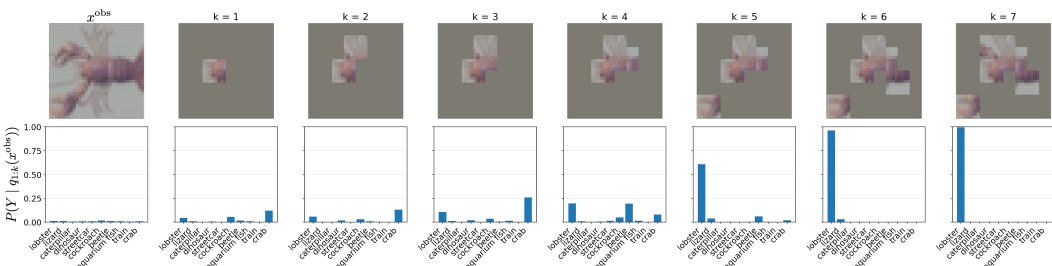

Figure 20: Examples for CIFAR100.

