# OpenReview forum: "Variational Information Pursuit for Interpretable Predictions"
_ICLR.cc/2023/Conference — ICLR 2023 poster_

### Official Review · Reviewer_G1eQ · 2022-10-20

**Confidence:** 3
**Correctness:** 4
**Technical Novelty And Significance:** 3
**Empirical Novelty And Significance:** 3
**Recommendation:** 8

**Clarity, Quality, Novelty And Reproducibility:**

The article is very clear and well written. It provides a new, solid and elegant reformulation of the problem of Information Pursuit.

The appendix provides (almost, see weakness 1 above) all necessary information to reproduce all the experiments in the paper.

**Strength And Weaknesses:**

**Strengths:**
- The proposed approach is an elegant and well-grounded variational reformulation of the IP strategy, which is itself a strong strategy in this context.
- The formulation of the optimization problem and its implementation using deep neural networks is clear and detailed.
- The experimental validation is extensive and discusses in depth the advantages and limitations of the proposed V-IP.
- The proposed model is competitive with previous state of the art with a significantly smaller computing cost, which is a strong contribution.

**Weaknesses:**
1. Regarding the straight-through optimization of the softmax of the querier model, as discussed in appendix D this requires the following function to be differentiable wrt to the one-hot output of the softmax. In V-IP, the following function is the concatenation of the next query result to the queried dataset. How is that step made differentiable? I suppose this is related to the masking-based architecture, but the paper does not seem to explain it.
2. This is a minor point, but I think figures 3a illustrates well how query sets consisting in observing small patches of an image are very artificial given the high-level problem at hand (interpretability). The 4th observation is the one that tips the balance of the model from cat to dog, and yet it mostly consists in a black square of background. Could that maybe illustrate some overfitting in the model?

**Summary Of The Paper:**

This articles positions itself in the context of designing "interpretable by design" models: models are built from the start to provide an interpretable explanation alongside their predictions. To achieve this goal, the model performs its prediction by iteratively making queries about the example to classify within a predefined set of interpretable queries. The sequence of queries serving as the explanation for the prediction. The paper follows the Information Pursuit (IP) method, where the next query is chosen by maximizing the mutual information between the variable to predict and the result of the query given the past query history.

Previous work in the IP framework learned a joint generative model over the query answers and the final prediction, using MCMC sampling on that model to compute the mutual information required to choose the next query. The paper shows that the IP selection strategy can be formulated as the solution to a variational optimization problem involving two functions: a classifier that predicts the output variable given the history of queries, and a querier, that chooses the next query. This formulation alleviates the need for learning a joint generative model, drastically reducing the computational cost.

The paper provides extensive experimental comparison with other models from the literature and on multiple tasks, proving the proposed V-IP model to be very competitive across the board.

**Summary Of The Review:**

This paper proposes a new method, which is well theoretically founded and empirically shows significant computational gains compared to the state of the art. There are a few details that can be clarified, but this is a good paper.

---

> ### Author Response · Authors · 2022-11-19
> **Response to Reviewer G1eQ**
>
> We thank the reviewer for the insightful feedback. We will address each comment below:
>
> **Comment 1: “Regarding the straight-through optimization of the softmax of the querier model, as discussed in
> appendix D this requires the following function to be differentiable w.r.t. to the one-hot output of the softmax. In
> V-IP, the following function is the concatenation of the next query result to the queried dataset. How is that step made
> differentiable? I suppose this is related to the masking-based architecture, but the paper does not seem to explain it.”**
>
> Thank you for pointing this out. In response, we have included a paragraph titled “Updating the history.” to the experiment details for each dataset explaining this in detail in Appendix C. To summarize for the masking-based architecture, the output of the querier model is a one-hot vector. This one-hot vector is point-wise multiplied to a vector of all query answers resulting in a vector v (assuming binary queries, more general dataset-specific cases in the appendix). We then add v to the current history (masked input based on observed query-answers so far) to get an updated history vector. This updated history is then passed onto the classifier network. Since every step mentioned here is differentiable, the classifier output is differentiable in terms of the one-hot output of the softmax (the querier model’s output).
>
>
> **Comment 2: “This is a minor point, but I think figures 3a illustrates well how query sets consisting in observing
> small patches of an image are very artificial given the high-level problem at hand (interpretability).”**
>
> We agree with the reviewer that query sets consisting of small patches are not ideal from an interpretability point of view. It does not tell us what the model is really looking at in those patches. We made this choice to compare with prior work that use patches (also called “glimpses” in those works). One major advantage of our method is that the end-user decides the query set. If one desires a more semantic interpretation than most informative regions in an image then the queries can be chosen accordingly. As an example in this paper we propose to solve the task of bird species identification by asking semantic queries about various high-level visual attributes of the bird like wingspan, beak shape, feather colour etc.
>
>
> **Comment 3: In Figure 3a., the 4th observation is the one that tips the balance of the model from cat to dog, and
> yet it mostly consists in a black square of background. Could that maybe illustrate some overfitting in the model?**
>
> We do not believe that this illustrates over-fitting since the fourth patch also reveals a small part of the
> dog’s ear (which can be inferred from context using the history of observations obtained from the first 3 patches).
> Cat’s typically do not have hanging ears.

---

> > ### Comment · Reviewer_G1eQ · 2022-11-21
> > **Thank you for your response**
> >
> > Thank you for your response which clarifies the uncertainties I had.

---

### Official Review · Reviewer_jPMz · 2022-10-23

**Confidence:** 4
**Correctness:** 4
**Technical Novelty And Significance:** 3
**Empirical Novelty And Significance:** 3
**Recommendation:** 8

**Clarity, Quality, Novelty And Reproducibility:**

Variational Information Pursuit for Interpretable Predictions:
Section 2: When you write "in almost all cases our method performs better." it would be good to add precision about what is ment to be better?

Section 3: you say that "algorithm terminates after $L$ queries if ... ", $L$ is here capitalized so it seems to indicate a user defined parameter. Would better be said that if
arg max I is 0 algorithm terminates. Please clarify.
- About stopping criterion in difference of entropies: ", which is difficult to compute without explicit generative modeling of the query-answer distribution". Could you clarify this last part, if it is hard to estimate then how you do it?

Section 4.2: please explain RAM and RAM+ at least with few words, maybe in a similar way as G-IP had been explained in earlier Sections. In Fig 4,  why RAM performance goes down in Huffington dataset?

In Fig 4. I would like to see baseline performance of some reasonable non-interpretable model that does not use queries but just takes the whole input as is and produces one output. It would be important to to see how much we lose by trying to be interpretable.

Minors:
- Please select to use either V-IP or VIP.

**Strength And Weaknesses:**

Positives:
- I find the paper very interesting and clearly written.
- Idea is neat and does seem to work in practice.
- I feel that in terms of quantitative performance it would be better to emphasize much more the speed of inference. Maybe even with some Figures, so that the message is really hammered home that V-IP is much faster than G-IP.
- Theoretical development is also a plus.

Negatives:
- I do not understand why in (V-IP) authors need to formulate their objective as a constraint optimization task. Why not to directly optimize f,g?
- Authors state in the Introduction that "We empirically demonstrate the efficacy of the proposed method over generative modelling on various computer vision and NLP tasks", but in the case of MNIST clearly V-IP is not the clear winner. I am thinking that authors should modify these statements to reflect the results more accurately.
- Thing that is sometimes hard to do is to put plot standard deviations in Figs and mark them into the results Tables. In RL lit, this is standard practice, you need to repeat the experiments some number of times to obtain those std estimates. Please consider adding them where ever possible.
- Clear limitation of the present paper is to have good Q set that allows for interpretation. If one constructs the set Q basically randomly, is it going to give useful information to the practitioner? It is on the other hand quite clear that if Q is very limited, then the proposed method would not work too well against non-interpretable models. Some discussions about this could be good to have.

**Summary Of The Paper:**

In this paper authors proposed a variational information pursuit for interpretable classification model / estimation scheme. Authors idea is motivated by generative variant that they denote G-IP. Authors propose a complete framework with model defintions explanation of the training scheme and proof that loss does what it is supposed to do. Finally they show with experiments how the proposed method works in contrast to baselines.

**Summary Of The Review:**

I find the ideas in the paper to be quite interesting and potentially very useful for practitioners. The main experimental results are not necessarily very compelling in terms of classifier accuracy, but in terms of inference speed they are.

---

> ### Author Response · Authors · 2022-11-19
> **Response to Reviewer jPMz (Part 1)**
>
> Thank you for the insightful feedback. We are happy that you find our work interesting. We address each of your
> response in the following:
>
> **Comment 1: I do not understand why in (V-IP) authors need to formulate their objective as a constraint optimization task. Why not to directly optimize $f$, $g$?**
>
> The V-IP objective in equation (V-IP) is not really a “constrained optimization problem” and we thank the reviewer for pointing this out. To remedy this confusion, we have now changed the wording from “subject to” to “where” in the relevant equations [(V-IP) and (Deep-VIP)].
>
>
> **Comment 2: Authors state in the Introduction that “We empirically demonstrate the efficacy of the proposed method over generative modelling on various computer vision and NLP tasks”, but in the case of MNIST clearly V-IP is not the clear winner. I am thinking that authors should modify these statements to reflect the results more accurately.**
>
> By efficacy we meant that empirically V-IP achieves competitive performance with the generative modelling approach but has huge speed gains, which is the intended result. In response to the reviewer’s critique we have now changed the sentence to the following:
>
> “Empirically, we show that V-IP achieves competitive performance with the generative modelling approach on various computer vision and NLP tasks with a much faster inference time.”
>
>
> **Comment 3: Please consider adding standard deviations in figures and results wherever possible.**
>
> Thank you for this suggestion. We have updated Figure 4, 5 and Table 2, 3 (in the main paper) with standard deviations wherever possible.
>
>
> **Comment 4: Clarity regarding “Section 2: When you write ‘in almost all cases our method performs better.’ it would be good to add precision about what is meant to be better?”**
>
> We apologize for the lack of clarity. This has been changed. The new writing in Section 2 reads as follows:
>
> “. . . we compare V-IP with prior works in this area and show that in almost all cases our method requires a smaller number of queries to achieve the same level of accuracy.”
>
>
> **Comment 5: Section 3: you say that “algorithm terminates after $L$ queries if ... ”, $L$ is here capitalized so it seems to indicate a user defined parameter. Would better be said that if $\arg\max I$ is 0 algorithm terminates. Please clarify.**
>
> We do not use the convention that capitalized letters denote user-defined parameters and hence $L$ is not user-defined and depends on the data-point $x^{obs}$. The algorithm terminates if all remaining queries are nearly uninformative (as measured by mutual information). The new text in the main paper now reads as follows,
>
> “The algorithm terminates after $L$ queries, where $L$ depends on the data point $x^{obs}$, if all remaining queries are nearly uninformative, that is, $∀q ∈ Q I(q(X); Y | q_{1:L})) ≈ 0.$”
>
>
> **Comment 6: About stopping criterion in difference of entropies: ”, which is difficult to compute without explicit generative modeling of the query-answer distribution”. Could you clarify this last part, if it is hard to estimate then how you do it?**
>
> We apologize for the lack of clarity. What we meant to say was that Chattopadhyay et al. (2022) used an information-theoretic criteria for stopping, which was that one would stop asking queries if the remaining unobserved queries have mutual information with the label $Y$ , given history of query-answer observed so far, lower than ε (an user-defined parameter). This requires explicit generative modeling of the joint distribution between query-answers and $Y$ . In V-IP however, since we do not learn an explicit generative model of the query-answers and $Y$ , we cannot use this information-theoretic stopping criteria directly. Instead, we use our learnt classifier to construct an unbiased stochastic estimate of this information-theoretic stopping criteria (assuming the classifier learns the distribution $P (Y | history)$ perfectly for any given history of query-answers) which we refer to as the “stability criterion” in this paper.

---

> ### Author Response · Authors · 2022-11-19
> **Response to Reviewer jPMz (Part 2)**
>
> **Comment 7: Section 4.2: please explain RAM and RAM+ at least with few words, maybe in a similar way as G-IP had been explained in earlier Sections.**
>
> We refer the reviewer to the description of RAM and RAM+ in the last 7 lines of the first paragraph
> (titled Baselines) in section 4.2.
>
>
> **Comment 8: In Fig 4, why RAM performance goes down in Huffington dataset?**
>
> In RL-based methods (RAM and RAM+), both the classifier and policy networks are trained in tandem to maximize the cumulative sum of rewards. In RAM, the agent obtains a reward (negative cross entropy between true and predicted labels) only at the last step. Thus, the classifier always sees the entire input while making a prediction during training. We conjecture that when the action space is large (such as $|Q| = 1000$ as in the case of the HuffingtonNews dataset), a classifier learnt on seeing the entire input fails to generalize to scenarios where only partial input is seen (as is the case when a few query answers are revealed). This is potentially why the performance in RAM initially goes down on the HuffingtonNews dataset. This phenomenon is not observed in RAM+, since the agent gets a reward equal to the negative cross entropy between true and predicted labels *at each step*. Thus, the classifier learns to make predictions based on partial inputs during training, that is, inputs resulting for masking based on history observed at any step.
>
>
> **Comment 9: Clear limitation of the present paper is to have good $Q$ set that allows for interpretation. If one constructs the set $Q$ basically randomly, is it going to give useful information to the practitioner? It is on the other hand quite clear that if $Q$ is very limited, then the proposed method would not work too well against non-interpretable models. Some discussions about this could be good to have. In Fig 4. I would like to see baseline performance of some reasonable non-interpretable model that does not use queries but just takes the whole input as is and produces one output. It would be important to to see how much we lose by trying to be interpretable.**
>
> Yes, an important aspect of the framework utilized in this work is that the end-user defines queries that are interpretable to them. Given this set of queries, $Q$, V-IP learns to efficiently compose them into concise explanations for model predictions (in terms of query-answer chains). The queries also need to be task and domain-dependent. If $Q$ is chosen randomly, there is no guarantee the queries would have any specific interpretation or even be sufficient for solving the task, that is, allow for making predictions with high accuracy. An illustration of this is provided in Chattopadhyay et al. (2022) Figure 2.
>
> It is also true that if $Q$ is limited, the proposed method will not work too well against non-interpretable models.
> We discuss this in detail in Appendix §G. In Table 5, in the appendix, we report performance using non-interpretable
> models that do not use queries. In almost all datasets, there is a drop in performance when basing predictions on
> interpretable query sets. This is expected since interpretability can be seen as a constraint on learning. Finding good
> query sets that are interpretable and allow for highly accurate predictions is a subject of future work.
>
>
> **Comment 10: Please select to use either V-IP or VIP.**
>
> Thank you for pointing out our typos. We have changed all instances of “VIP” to “V-IP”.

---

### Official Review · Reviewer_VaCA · 2022-11-04

**Confidence:** 3
**Correctness:** 3
**Technical Novelty And Significance:** 2
**Empirical Novelty And Significance:** 3
**Recommendation:** 6

**Clarity, Quality, Novelty And Reproducibility:**

# Clarity

I enjoy reading the paper and it's quite smooth and clear. One little thins is maybe authors can illustrate that those queries are in fact the features in a dataset in Fig. 1. Although I understand the queries are more general, it makes me confused that if an oracle is needed to get the answer of these questions or it needs to be learned. I realize that those questions in Fig. 1 are in fact just a feature value in the dataset in the experiment sections so no oracle is needed.

# Quality
The experiments are overall good. One important thing is can authors please add standard deviation on both Table 2, 3 and Figure 4, 5 to understand the significance of each method?

# Originiality
IMHO, I think the work has slightly lower originiality since these methods are similar to RL-based methods.
1. It reminds me of an earlier work [1] that also uses RL to do per-instance feature selection. Can authors please comment on the relations?
2. Can authors comment on in what scenarios the proposed greedy approach work better (gamma=0), and in what scenarios the RL-based approaches (gamma > 0) can be better? [1] seems to show that the RL-based approaches perform better.

# Thoughts (may not be important)
1. Can this method be further improved by combining with a generative approach such as partial VAE? For example, a way to improve V-IP is that for each feature selection doing an imputation for the rest of unselected features and use all the features to send to the classifier. I think it will improve in the image space where there is a high-degree of correlations. Do you think if this approach will improve the accuracy, and also the interpretability?


[1] INVASE: Instance-wise Variable Selection using Neural Networks: https://openreview.net/forum?id=BJg_roAcK7

**Strength And Weaknesses:**

# Strength

- The writing is clear and easy to follow.
- The examples of interpretability shown in Fig. 3 are interesting to see.
- I like the careful details of biased samplings and the set-based classifiers v.s. mask-based classifiers.
- The experiments seem thorough, but some more experiments can be helpful. See below.

# Weaknesses

1. IMHO, the novelty side may be a little bit low since this method can be seen as a RL method with immediate reward of improving the classifiers predictions. Other inventions like initial random sampling and subsequent biased sampling are new in my opinions.
2. G-IP seems to perform quite similarly or sometimes better than V-IP (Fig. 4) in larger datasets like CUB-200 with large query size 312. It seems contradictory to what authors state "We thus expect the gains of V-IP to be most evident on large-scale datasets". An ablation study that improves from small to large number of examples in a dataset may help verify what the authors claim.
3. There is no consistent baselines across the datasets, which may make comparisons difficult. If it's easy to do, can authors also run the G-IP on the datasets in Table 3 and Figure 5 CIFAR-10/CIFAR-100 to see if the proposed method V-IP is indeed better than G-IP? Or authors can comment on why such comparisons are not easy due to code inaccessibility etc.
4. I would love to see a more complete ablation study than the Supp. E (Fig. 11) that compare another version <1> No initial random sapling, and put a subset of results into a small table in the main text if possible.
5. The metrics reported in the experimental section are mostly borrowed from other papers. Although it's great to directly compare to the SOTA numbers, there maybe subtle differences in the implementation leading to such result. For example, can authors please confirm fi those baselines BSODA, REFUEL in Table 3 and Figure 5 are using the same classifier architectures and training strategies the same as V-IP and the only difference lies in the strategy of selecting the features? If not, such number should not be directly compared, or some ablation studies are needed e.g. using BSODA's classifiers instead of the V-IP classifiers.

**Summary Of The Paper:**

This paper proposes a method called V-IP (Variational Information Pursuit) that does a multi-step prediction to improve interpretability instead of doing a one-pass prediction like other neural nets do. In each step, only a small sets of features (i.e. "query set" called in the paper) are revealed and the goal is to make a prediction using the minimun number of steps i.e. part of the feature sets. It can derive interpretability becasue the subsets of features causing a big increase of the prediction of the ground truth class between steps can be seen as important rationales of why model makes such prediction.

Previously, most method resort to using generative models to model the distributions between labels and subsets of features to pick which parts of features can maximally predict the target by resorting to MCMC sampling methods (the baseline called G-IP). Or others have proposed using reinforcement learning to sequentially select the feature sets that predict the correct target. The proposed method, V-IP, instead learns to greedily choose the subsets of features that maximize the downstream classifiers to predict the target y in each step as measured in the KL divergence, which can be seen as the mutual information of the current selected features in each step. Note V-IP can be seen as a RL method that has an immediate reward and the decay factor gamma set to 0. In the classification part, V-IP experiments with set-based and mask-based classifiers to predict and find the mask-based ones perform better. In a wide-variety of datasets including images, medical diagnosises, and texts data, the V-IP outperforms recent G-IP related methods and RL-based methods.

**Summary Of The Review:**

Overall I think the authors do a great job in writing and the examples presented are interesting to see. But given the crowded space of this problem (feature subsets selection), I am hoping to see more complete comparisons and other interesting questions as suggested above. The experimental number also lacks standard deviation which I believe is important. Although I find this work slightly less original, I don't mind such work being accepted as long as there are concrete evaluations and ablation studies to make readers learn something from it.

---

> ### Author Response · Authors · 2022-11-19
> **Response to Reviewer VaCA (Part 1)**
>
> We thank the review for the insightful feedback. We will address each of your comment in the following:
>
> **Comment 1: “Novelty side may be a little bit low since this method can be seen as a RL method with immediate
> reward of improving the classifiers prediction.”**
>
> While we agree with the reviewer that our V-IP objective can be seen as a myopic RL agent with immediate rewards (with a discount factor γ = 0), we respectfully disagree that this reduces the novelty of this work. First, our work makes an interesting and unexpected theoretical connection that, to the best of our knowledge, has not been reported in the (RL) literature. Specifically, we theoretically prove that minimizing our proposed KL-divergence based VIP objective is akin to finding the most informative query. This motivates using VIP to learn the most informative query directly instead of first learning a generative model to estimate mutual information. Second, we propose to optimize our objective using the straight-through softmax estimator, which is empirically known to have much lower variance than the commonly employed policy gradients in RL [1,2]. This makes the implementation for V-IP simpler than policy-gradient based RL algorithms since we do not need to train an auxiliary baseline network to reduce the variance of the estimated gradients.
>
> [1] Paulus, Max B., Chris J. Maddison, and Andreas Krause. "Rao-blackwellizing the straight-through gumbel-softmax gradient estimator." arXiv preprint arXiv:2010.04838 (2020).
>
> [2] Chen, Jianbo, Le Song, Martin Wainwright, and Michael Jordan. "Learning to explain: An information-theoretic perspective on model interpretation." In International Conference on Machine Learning, pp. 883-892. PMLR, 2018.
>
> **Comment 2: G-IP seems to perform quite similarly or sometimes better than V-IP (Fig. 4) in larger datasets like CUB-200 with large query size 312. It seems contradictory to what authors state ”We thus expect the gains of V-IP to be most evident on large-scale datasets”. An ablation study that improves from small to large number of examples in a dataset may help verify what the authors claim.**
>
> There is no contradiction in our statement. By large-scale datasets we meant datasets with a large number of training points and not the size the of query set. CUB-200 has only about 5000 training points, which is much smaller than the datasets where VIP outperforms the generative approach like CIFAR-{10,100} which have about 50k training points, or the SymCAT datasets which have about a million datapoints.
>
> That being said, the reviewer’s comment suggests we need to better explain what we meant to say: If the distribution P (Q(X), Y ) can be learned well by a generative model that allows for tractable inference, then we expect G-IP to perform better since it computes the most informative query by explicitly computing the required mutual information terms. We expect the gains of V-IP to be more evident on datasets where learning the generative model is more difficult (for example, we show V-IP performs much better than the generative approach on RGB-CIFAR images but lags behind on the MNIST dataset where good generative models are available). In light of this confusion, we have changed our original statement to the following in the main paper:
>
> “We conjecture that, when the data distribution agrees with the modelling assumptions made by the generative model, for example, conditional independence of query answers given $Y$ , and the dataset size is “small,” then G-IP would obtain better results than V-IP since there are not enough datapoints for learning competitive querier and classifier networks. We thus expect the gains of V-IP to be most evident on datasets where learning a good generative model is difficult.”

---

> ### Author Response · Authors · 2022-11-19
> **Response to Reviewer VaCA (Part 2)**
>
> **Comment 3: There is no consistent baselines across the datasets, which may make comparisons difficult. If it is easy to do, can authors also run the G-IP on the datasets in Table 3 and Figure 5 CIFAR-10/CIFAR-100 to see if the proposed method V-IP is indeed better than G-IP? Or authors can comment on why such comparisons are not easy due to code inaccessibility etc.**
>
> We disagree with the reviewer. This work proposes a variational characterization of Information Pursuit (IP) which does away with generative models and tries to directly optimize a KL-divergence-based objective to find the most informative query, as required by IP, in each iteration. Across all datasets, we compare V-IP’s performance with the generative approach to IP, which first proposes to learn an explicit generative model for query answers and labels and then uses this model to estimate mutual information for every query in order to select the most informative query in each iteration. From this viewpoint, we believe our baselines for the generative approach to IP are consistent across datasets. The three algorithms G-IP, Probabilistic Hard Attn and BSODA only differ in the generative model used to learn the distribution of query answers and labels. All three methods still perform information pursuit by using this generative model. Having said so, the reason why we did not compare with G-IP for all datasets in the original paper is that we found that the generative model it uses is too limiting for complex tasks. Specifically, we learnt a VAE for the CIFAR-10 dataset (using an off-the-shelf VAE implementation) and tried to carry out IP with it (as dictated by the G-IP algorithm). However, we found that the model only achieves a paltry 50.22% test accuracy upon seeing all the query answers (which is an upper bound to the accuracy we can get with IP which is based on predictions made from partial observations of query answers). Thus, we moved on to better generative models introduced in the BSODA and Probabilistic Hard Attn. papers for CIFAR-{10,100} and the medical diagnosis datasets.
>
>
> **Comment 4: About a more complete ablation study, and comparisons on with initial random sampling versus no initial random sampling.**
>
> We thank the reviewer for this suggestion and have carried out this ablation study as requested. The results are reported in Figure 11 in the Appendix. Due to the large amount of experiments requested (Ablation studies for 12 datasets + multiple runs for std of curves) we were unable to complete this additional ablation for the CIFAR-10 dataset. However, we expect the trend to stay the same and would report this in the final version of the paper.
>
> We observe that in most datasets, biased sampling without the initial random sampling phase ultimately ends up learning a slightly better strategy (in terms of accuracy vs. explanation length trade-offs). However, since biased sampling requires multiple forward passes through a neural network to generate histories, it is much slower than the initial random sampling. Thus, when computational resources are not a concern one could do away with the initial random sampling but under a computational budget, the random sampling allows for finding a quick solution which can then be finetuned using biased sampling. This finetuning will potentially require fewer epochs to get good performance than training using biased sampling from scratch.

---

> ### Author Response · Authors · 2022-11-19
> **Response to Reviewer VaCA (Part 3)**
>
> **Comment 5: The metrics reported in the experimental section are mostly borrowed from other papers. Although it is great to directly compare to the SOTA numbers, there maybe subtle differences in the implementation leading to such result. For example, can authors please confirm if those baselines BSODA, REFUEL in Table 3 and Figure 5 are using the same classifier architectures and training strategies the same as V-IP and the only difference lies in the strategy of selecting the features? If not, such number should not be directly compared, or some ablation studies are needed e.g. using BSODA’s classifiers instead of the V-IP classifiers.**
>
> The results in Table 3 and Figure 5 show that our proposed approach (V-IP) performs much better than BSODA and REFUEL algorithms on most datasets in terms of higher accuracy achieved using the same number of queries. It is very difficult to evaluate the query-selection strategy (the querier) in isolation from the classifier. The reason is that the classifier is designed to operate on the data selected by the querier, thus the optimal classifier changes if we change the querier. More specifically, BSODA’s classifiers are trained on complete observations (all query answers) and a partial-VAE is trained to complete partially observed histories. Thus, the performance of VIP’s strategy using BSODA’s classifier would implicitly depend on the quality of this learnt partial-VAE which goes against our claim “we expect V-IP to work better on tasks where learning good generative models is difficult”. Similarly, REFUEL is an RL method and its classifier is trained to achieve good performance on only the query-chain histories its policy observes during training. Evaluating REFEUL’s classifier on V-IP’s query strategy would most likely result in sub-optimal performance.
>
> Unfortunately, both REFUEL and BSODA do not reveal their exact architectural details. The BSODA paper gives a high-level mathematical description of the set architecture used for their partial-VAE networks. We use the same set architecture for our querier and classifier networks (Appendix §C.4), but the exact details like the number of layers and the number of neurons in a layer might vary. Moreover, to get good performance, both BSODA and REFUEL employ multiple approximations and/or auxiliary losses. We used none of these in training our V-IP model.
>
> **Comment 6: Clarification: Are the queries just feature values in a dataset so no oracle is needed?**
>
> In most datasets considered, the queries are features in the dataset. However, in the CUB-200 dataset
> (which is the basis for the example in Figure 1), we train CNNs to answer each of the queries and hence can be
> seen as IP with noisy answers. These CNNs are trained using the attribute annotations present in the dataset. This is
> mentioned in Appendix §C.1 where we describe the task and dataset in more detail. We train our V-IP querier and
> classifier networks directly on these noisy answers provided by the learnt CNNs so the noise is taken into account
> while learning good strategies.
>
>
> **Comment 7: “One important thing is can authors please add standard deviation on both Table 2, 3 and Figure 4,
> 5 to understand the significance of each method.”**
>
> We have updated most figures and tables to reflect the mean and standard deviation of the performance over five runs, using a different seed each time. Since G-IP is extremely compute intensive (few weeks to run inference on the MNIST test set), we are unable to do multiple runs for G-IP.

---

> ### Author Response · Authors · 2022-11-19
> **Response to Reviewer VaCA (Part 4)**
>
> **Comment 8: IMHO, I think the work has slightly lower originality since these methods are similar to RL-based methods. It reminds me of an earlier work INVASE[3] that also uses RL to do per-instance feature selection. Can authors please comment on the relations?**
>
> We disagree with the reviewer’s comment. While feature subset-selection method like INVASE[3] could potentially be used to make predictions interpretable in terms of finding a small set of features that are most relevant for prediction, it does not explain the underlying decision-making process/reasoning behind the model’s predictions. In sharp contrast, the proposed V-IP framework, sequentially selects queries (which are user-defined interpretable features/functions) one at a time in order of information-gain. At each step, the selection of the next query is a deterministic function of the history of query-answers observed so far. Moreover, the user also has access to how the model’s posterior belief over the different labels change over time as more and more evidence is accumulated from query-answer histories. Such rich description of the model’s decision-making process is not possible by subset-selection methods like INVASE[3] which given an input $x^{obs}$, simply selects a small set of features and then makes a prediction based on these features.
>
> [3] Yoon, Jinsung, James Jordon, and Mihaela van der Schaar. "INVASE: Instance-wise variable selection using neural networks." In International Conference on Learning Representations. 2018.
>
>
> **Comment 9: “Can authors comment on in what scenarios the proposed greedy approach work better ($\gamma=0$), and in what scenarios the RL-based approaches ($\gamma > 0$) can be better? [3] seems to show that the RL-based approaches perform better”**
>
> We are unsure what the Reviewer means by “[3] show that the RL-based approach performs better”. [3] is not a sequential-decision making problem and there is no Markov Decision Process defined with an agent trying to maximize some discounted reward, for the γ parameter in question to be relevant in [3].
>
> In this paper, through experiments on numerous datasets across multiple domains (Vision, NLP, Medical Diagnoses) we show that for the particular problem of sequential decision making for interpretable predictions, the greedy strategy
> of choosing the most informative query (corresponding to $\gamma = 0$) outperforms popular RL algorithms from literature that can be adapted for solving this problem (namely, RAM and RAM+). We believe a detailed empirical and theoretical study is needed before drawing more general conclusions as to when $\gamma > 0$ would be preferred. This however is out of the scope of the current paper which is about interpretability.
>
>
> **Comment 10: “Can this method be further improved by combining with a generative approach such as partial VAE? For example, a way to improve V-IP is that for each feature selection doing an imputation for the rest of unselected features and use all the features to send to the classifier. I think it will improve in the image space where there is a high-degree of correlations. Do you think if this approach will improve the accuracy, and also the interpretability?”**
>
> We thank the reviewer for this suggestion, and a hybrid approach might be worth exploring in the future. Having said so, we observed in our experiments with the medical diagnosis datasets that the approach suggested by the reviewer, which is exactly what BSODA does, performs inferior to V-IP on the SymCAT datasets in terms of accuracy achieved using the same number of queries.

---

### Author Response · Authors · 2022-11-19
**Summary of Changes in Rebuttal Submission**

We are pleased the reviewers found our work interesting, well-written and potentially useful to practitioners (Reviewer jPMz) and thank them for their time. In response to the reviewers’ comments, we have made several changes to the revised rebuttal version. These changes are annotated in blue. We highlight some of our modifications in the following:

1. As requested by Reviewer VaCA and jPMz, we repeated most of the experiments in Table 2, 3 and Figure 4 and 5b, five times with different random seeds and reported standard deviation values.

2. As requested by Reviewer VaCA, we conducted a further ablation study of training with only the biased sampling scheme and reported results in Figure 12 in the Appendix.

3. As requested by Reviewer jPMz, we added a discussion in Appendix §G about limitations of the present paper in terms of having to select a good query set that allows for interpretation. We also added results in Table 5 of the Appendix, comparing classification accuracies of non-interpretable models (for datasets used in this paper) with the performance V-IP obtains using interpretable query sets.

4. As requested by Reviewer G1eQ, we added details in the appendix (for every dataset) as to how the query selection and concatenating it to history is made differentiable in §C.

---

### Decision · Program_Chairs · 2023-01-20

**Decision:**

Accept: poster

**Justification For Why Not Higher Score:**

I'd say this should be either a poster or a spotlight.  This could be a spotlight and I wouldn't argue against it.  Interpretable predictions for deep networks seems like something that is of general interest to the community.  I didn't choose spotlight because one reviewer gave a 6.

**Justification For Why Not Lower Score:**

This is a clear accept.  It has a high review score, is timely, novel and should be of interest to the community.

**Metareview: Summary, Strengths And Weaknesses:**

In this work, the authors develop a method to efficiently create predictive algorithms for deep networks that are interpretable by design, through a novel variational approach to Information Pursuit.  This involves asking interpretable queries about the data, choosing queries based on an information gain criterion.  The authors develop a variational version of IP that is much more efficient than previous versions that required MCMC with an additional generative model.  Across a variety of experiments the authors demonstrate that the variational approach is highly competitive while significantly more efficient than the generative model baseline.

The reviewers found the work clear, well written, technically correct and the experiments exhaustive.  One reviewer found that the work was not particularly novel when considered as an RL method with immediate reward.  The other reviewers seemed to find it novel, however.  In general, the reviewers found the setting quite compelling, i.e. interpretable by design deep network predictions seems timely, important and interesting to the community.  Overall the reviewers voted for an accept (6, 8, 8).  Therefore, the recommendation is to accept the paper.

**Note From Pc:**

if the above contains the word "oral" or "spotlight" please see: "oral" presentation means -> notable-top-5% and "spotlight" means -> notable-top-25%. As stated in our emails, we are disassociating presentation type from AC recommendations